# Game Theory-Based Load-Balancing Algorithms for Small Cells Wireless Backhaul Connections †

**Zsolt Alfred Polgar * and Mihaly Varga**

Communications Department, Technical University of Cluj Napoca, 400114 Cluj Napoca, Romania
* Correspondence: zsolt.polgar@com.utcluj.ro; Tel.: +40-264401226
† This paper is an extended version of our paper published at the 2022 45th International Conference on Telecommunications and Signal Processing (TSP).

**Featured Application: The proposed algorithms could be integrated into mobile router platforms installed in public transportation vehicles which can provide high-speed internet connectivity to the passengers and the equipment of the vehicles. The proposed algorithms also could be integrated into Access Points (APs) serving (possibly temporarily installed) small cells, APs that do not have dedicated backhaul connections, and the wireless links of some macro cells being used as backhaul connections.**

**Abstract:** 5G wireless networks have as one of the main characteristics the large-scale deployment of small cells (microcells, picocells, etc.), which is expected to bring several advantages in what concerns the high speed and low latency connectivity of the users. This large-scale deployment of small cells also raises several technical challenges, provisioning the backhaul connectivity being one of them. The paper considers the situations when small cells are deployed temporarily or are deployed in a vehicle transporting many passengers, situations when the traditional wired or wireless backhaul solutions could be too costly to be used. The paper proposes, as an alternative solution, the use as backhaul connections of the wireless links set up in the macro cells which cover the location of the small cell. The paper proposes several Game Theory (GT)-based Load-Balancing (LB) algorithms for distributing the traffic of the small cell users over the macro cell links. The proposed LB algorithms are evaluated by computer simulations and are compared with "classical" LB algorithms considered as references. The performed computer simulations show that the auction-based algorithms have the best performance in terms of delay suffered by the transmitted data packets, while the selfish routing type algorithm has weaker performance, even behaving poorly than some of the reference non-GT-based algorithms. The paper also considers the situation when several small cell APs are deployed in a limited area or a vehicle and the user groups that attach to different APs should be identified. The paper proposes two GT-based user clustering algorithms, and the performance of these algorithms are evaluated by computer simulations. These simulations show that even simple clustering algorithms could improve the distribution of the traffic over the neighbor small cell APs and reduce the delay experienced by the data packets in the transmission system.

**Keywords:** game theory; heterogeneous network; load balancing; packet delay; virtual tunnels

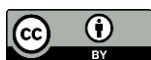

## 1. Introduction

Significant increases in mobile broadband speed, connectivity for a massive number of smart devices, low latency and high reliability wireless transmissions, energy efficiency, etc., are some of the improvements brought by the 5G networks [1,2]. To fulfill the requirements mentioned above, the designers of the 5G networks propose, besides the development of new Radio Access Technologies (RATs), a new network architecture that allows the integration of heterogenous RATs and the large-scale deployment of small cells [2,3]. The deployment of many small cells raises several challenges, one of them being the

provisioning of backhaul connections with large data rates, low latency, and high-reliability requirements [2]. Optical fiber links, the best solution for backhaul connections in 4G and 5G networks, might be too expensive or even impossible to be deployed in a vast number of locations and, as a viable alternative, the use of wireless backhaul is proposed [2,4]. The cost incurred by the deployment of the backhaul connections is a limiting factor of the "network densification" [5], having a significant influence on the number of small cells which can be deployed. In [6] it is presented a solution for optimization of the small cells' deployment process according to the type and characteristics of the backhaul connections. In some circumstances, such as small cells deployed in rural areas or small cells with low traffic or offering free access, wireless mesh networks are proposed to be used as backhaul technology, or even self-backhauling in mobile frequency bands. Due to the scarcity of the spectrum used for mobile access the last solution is useful only in particular circumstances. An analysis of the usage of mesh WiFi 802.11n networks for provisioning the backhaul connections for picocells is presented in [7], and a self-optimizing partial mesh wireless backhaul solution is proposed in [8].

The architecture of the 5G networks is a complex one organized on several functional layers and is based to a large extent on technologies such as Software Defined Networks (SDN) and network function virtualization [2,9]. This could allow the use of more complex data transmission mechanisms, such as LB over several heterogeneous communication links. LB allows more efficient usage of the available resources in a heterogeneous networks scenario and this concept has already been extensively studied in the context of WiFi-3G/WiMAX/4G heterogeneous wireless networks. In [10] the authors investigate the issue of parallel transmissions over multiple RATs, focusing their attention on the Quality of Service (QoS) perceived by the final users. A simple but efficient LB algorithm is proposed and evaluated in WiFi-UMTS and mobile WiMAX systems scenarios. In [11] an improved mechanism for routing the Internet traffic over several communication paths in heterogeneous networks is proposed. The heterogeneous network is represented as a combination of multiple single technology networks available simultaneously, and each network is modeled as a single server queue. The number of jobs in each queue is minimized subject to constraints using the Lagrange multiplier. A soft LB mechanism that divides the traffic of the users into sub-flows and routes these sub-flows through different wireless access networks is presented in [12]. Soft LB involves the combined use of load distribution and vertical handover techniques. In [13] the authors adopt a fuzzy neural network approach to determine the optimal load sharing of the traffic among the heterogeneous networks and in [14] heterogeneous wireless networks are mapped to distributed grids. The authors of [14] present a hierarchical semi-centralized architecture for balancing the traffic among heterogeneous wireless networks and the proposed solution is based on the grid concept in data transmission networks. The LB mechanisms in networks involving the WiMAX and WLAN technologies are investigated in [15]. The authors consider the different service characteristics of these two networks and the QoS requirements of real-time and best effort applications and distribute all streaming flows to WiMAX networks, the remaining capacities of WiMAX and the entire capacity of WLAN being used for transmission of flows associated with non-real-time services.

The advantages of LB over heterogeneous wireless links in what concerns the resource utilization, system capacity, and fulfillment of QoS requirements is proved in [16]. The authors propose a LB algorithm that is acting based on the scheduling mechanism used and the wireless link quality information. The proposed solution is more time effective and can operate in a distributed way. In [17] a new approach is proposed to compute a metric expressing the network load. The new metric computation hides the heterogeneity of network technologies from the LB module and can be applied to any packet injected into the system. In [18] a utility function-based approach, which can support multiple client classes is introduced. A bandwidth sharing policy in heterogeneous networks and a "controlled unfairness" scheme are achieved by using logarithmic utility functions that characterize the bandwidth allocated to different users.

In [19] a Markov decision process is proposed for LB operations in heterogeneous cellular networks using multiple carriers, and the LB solution proposed is based on the usage of the Sarsa algorithm of online learning. The authors of [20] jointly address the network LB and the server LB and propose a two-phase algorithm that concurrently deals with the two LB problems in the context of new network architectures integrating virtualized network functions. In [21] the authors present a LB algorithm that selects a suitable network interface for each data flow in a dynamic way. The paper considers the growing importance and the heterogeneity of Local Area Networks (LAN) and the increasing requirements of the services in terms of latency and throughput. The interface selection is based on service requirements and the LB operations are modeled as a mixed integer linear program.

In [22] the LB technique is proposed to be implemented among collocated BSs for better usage of the transmission resources and increased energy efficiency in the context of sustainable green wireless networks. In [23] the authors propose solutions to solve the problem of resource allocation optimization in heterogeneous 5G networks while satisfying the QoS requirements of the users. A multi-agent system is embedded into the standard cuckoo algorithm and the multi-agent cuckoo algorithm is obtained and used to solve the considered resource allocation problem. In [24] the authors analyze, using computer simulation, different scheduling schemes for radio resources in heterogeneous networks that integrate 5G, 4G, Wi-Fi, and other wireless technologies. The goal of the study is to determine how to improve the throughput in heterogeneous 5G networks.

Game theory and genetic algorithm-based LB mechanisms were also proposed in several papers. In [25] the authors study the bandwidth aggregation on backhaul connections in the context of wireless LANs composed of APs with spare backhaul capacity and APs with a shortage of backhaul capacity. The transfer of transmission capacity between the two categories of backhaul connection is modeled as a matching game with the many-to-one setting. In [26], considering the high demand for computation and communication resources and the heterogeneity of these resources in distributed computer systems, the authors propose a new method for solving the LB problem using game theory and a genetic algorithm. The LB problem is modeled as a non-cooperative game of the system's users, and a genetic algorithm is used to solve the formulated LB game. In [27] the authors consider the problem of mobile users' access to cloud services. To ensure high performance and reliability the integration of centralized cloud computing and distributed edge computing is considered. To meet the latency requirements of some computation-intensive applications the authors propose to divert some of the incoming job requests from overloaded cloudlets to under-loaded neighbor cloudlets and this LB process is modeled as a non-cooperative game. The problem of LB in data centers in the context of cloud computing is considered in [28]. The increase in the number of IoT devices and the increasing interest in real-time analytics services could generate significant imbalances in the load of cloud data centers. The paper proposes a Stackelberg (leader-follower) game model for selecting in a balanced way the physical host for executing each task arriving at the data center. A similar problem of LB in data centers in the context of cloud computing is also considered in [29]. The authors propose two solutions for the LB problem, considered a non-cooperative game among users, based on game theory and metaheuristic algorithms. The authors of [30] consider the problem of LB in distributed systems at a more general level. The LB problem defined has two conflicting objectives: minimize the users' expected response time and minimize the total cost incurred by the users. The authors model the LB problem as a non-cooperative game and propose an algorithm to solve the defined game. Offloading traffic from crowded base stations to APs in a 5G environment is considered in [31]. The authors propose a scheme using game theory and the Stackelberg approach for taking the decisions of traffic offloading of crowded networks. The proposed solution can achieve a more efficient QoS in a networking environment involving various types of data with different QoS requirements. The use of GT concepts in LB algorithms over virtual communications tunnels established between an AP node, which receives the

user flows, and an Advanced Gateway node is considered in [32]. The paper proposes several GT-based LB algorithms and compares the performance of these algorithms with that of "classical" LB algorithms, being demonstrated that some of the GT-based algorithms can manage most efficiently the transmission resources available in the virtual communication tunnels. The short state of the art presented above shows the use at large scale of GT-based algorithms in solving various load balancing and traffic offloading problems. This is mainly due to the flexibility of GT in modeling various optimization problems and the ability of this class of algorithms to solve efficiently the mentioned optimization problems.

This paper considers the situation of small cells deployment in an evolved wireless network when due to different technical and economic reasons the backhaul connection of the evolved small cell is implemented using the shared links of the macro cells covering the small cell (i.e., in-band and/or out-band self-backhauling). To provide the backhaul capacity required by the evolved small cell in discussion the aggregation of several macro cell links is considered, a process that requires splitting the traffic over the available links modeled as communication tunnels, which is a resource usage optimization problem, while specific constraints are imposed. Considering the ability of the GT-based algorithms of solving optimization problems the paper proposes several GT-based LB algorithms over the macro cell links providing the backhaul connectivity. The proposed algorithms are adapted to the specific case of LB over virtual communication tunnels with specific definitions of the games and specific QoS parameters which should be ensured. The performance of the proposed GT-based LB algorithms and of some reference LB algorithms are assessed by extensive computer simulations. The obtained results show the superiority of LB algorithms which model the LB process as a multiplayer game based on a sealed-bid auction process compared to non-GT-based algorithms and non-cooperative GT-based algorithms. The paper also considers scenarios when several small cell APs are deployed and proposes GT-based solutions for user clustering, i.e., the selection of the user groups that connect to each AP. The goal is the LB between the deployed APs to ensure the QoS requirements on the AP–user links, and of course to ensure the global end-to-end QoS.

## 2. Materials and Methods

### 2.1. Problem Formulation and System Model

The considered networking scenario is presented in Figure 1. The AP of the small cell offers high bit rate communications to its active users, using some access technology such as LTE, WiFi, etc., and connects to the core network using the wireless links of several macro Radio Access Networks (RANs) (LTE, 3G+, WiMAX, 5G, etc.) which cover the area of the small cell.

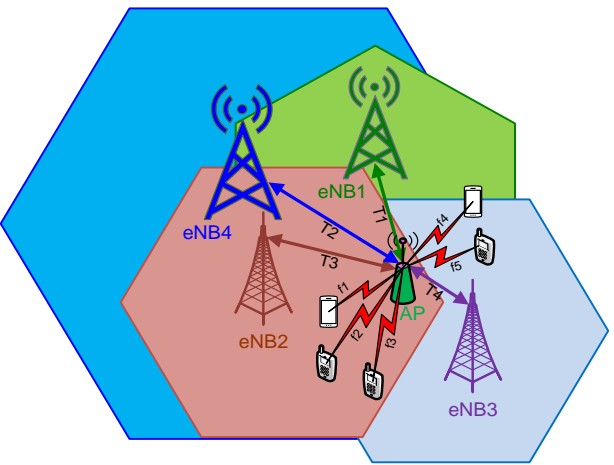

**Figure 1.** The considered networking scenario.

A particular case is represented by the deployment of a small cell inside a public transportation vehicle (bus, train, etc.) to provide ubiquitous and high bit-rate communications to the passengers and the equipment of the vehicles (video surveillance system, infotainment system, etc.). In this situation, depicted in Figure 2, one of the most cost-effective solutions for providing backhaul connectivity for the AP is to use the wireless links of the macro cells which cover the route of the vehicle as backhaul links. The AP is represented in this case by an Advanced Mobile Router which works together with a dedicated advanced gateway, the Service Continuity Gateway (see Figure 2), to provide ubiquitous connectivity.

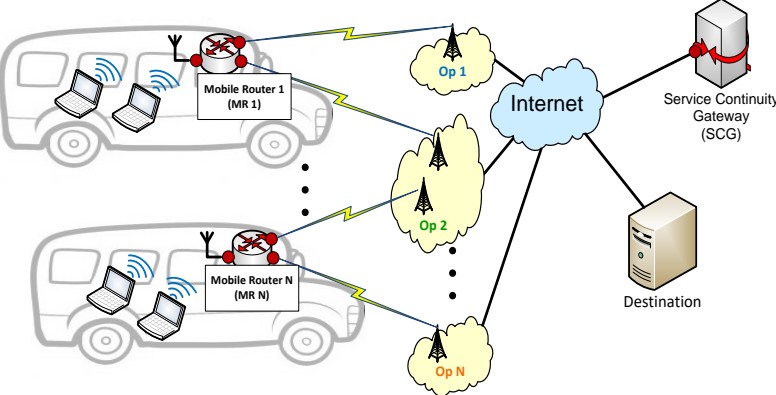

**Figure 2.** Heterogeneous network architecture that provides ubiquitous connectivity to public transportation vehicles.

A single shared macro cell link might not be able to provide the needed backhaul capacity, at least not in circumstances when the number of small cell users is large and bandwidth-demanding services are accessed. The solution is to use several macro cell links established in different macro cells covering the small cell and aggregate the bandwidth offered by these macro cell links. The problem which must be solved is how to distribute/route in an efficient way the service flows of the users connected to the small cell AP over the macro cells shared links while fulfilling the QoS requirements of the user's service flows.

The considered system model for the uplink transmission is depicted in Figure 3 and includes the AP node, which receives the services flows, $f_i$, generated by the users, and is connected to an Advanced Gateway node (AG) by several parallel virtual tunnels. Both nodes have LB capabilities over the virtual tunnels connecting them. Such an advanced gateway node could be part of a virtualized infrastructure layer [1,2], and its functionalities can be defined and implemented using SDN technologies [9]. As in most communication systems, e.g., LTE and WiMAX, each duplex communication channel is formed of two independent simplex communication channels. The modeling of the downlink transmission is like that of the uplink transmission, only the direction of the flows is different. Each flow $f_i$, $i = 1,…,K$, is characterized by a distribution function $\Gamma_i(t)$ which describes the packets' arrival rate, and a function $\Theta_i(x)$ which describes the distribution of the packets' size, $x$. The average bit rate $R_i^{av}$ of the $i$-th flow is given by:

$$R_i^{av} = \iint \Gamma_i(t)\Theta_i(x)dxdt \tag{1}$$

It is assumed that there are also several output flows and there is a one-to-one correspondence between the $i$-th input flow and $i$-th output flow.

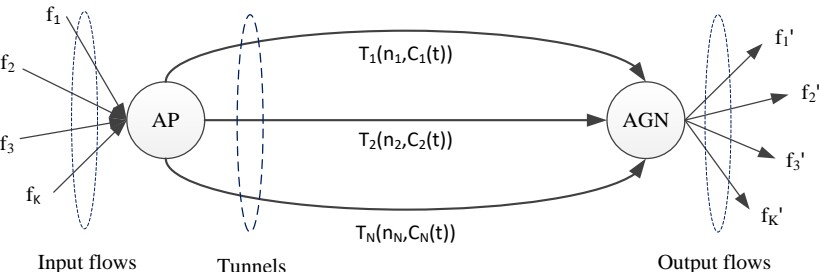

**Figure 3.** The system model of the considered networking scenario.

### 2.1.1. Virtual Tunnel Modeling

Each tunnel between the AP and AGN nodes is modeled as a First-In-First-Out (FIFO) queue with a single server whose service time depends on the instantaneous physical channel conditions in the wireless network where the tunnel is passing. To each tunnel, $j$, is associated a convex function $T_j(n_j, C_j(t))$, which returns the expected waiting time for a newly arrived packet, if, after queuing the packet there are $n_j$ packets waiting in the queue, and the instantaneous capacity of the physical link is $C_j(t)$.

Because the data flow fed to the input of a tunnel is a mixture of different types of traffic, real-time and non-real-time traffic with different packet sizes and packet interarrival times, it can be considered that the packets arrive according to a Poisson process [33]. It should be mentioned that the Poisson traffic model is not the most realistic model for many traffic types, but it is widely accepted and provides mathematical tractability. Some more complex modeling involving a mixture of Poisson distributions is given in [33]. The data packets' interarrival times on different tunnels are independent random variables exponentially distributed with parameter $\lambda_{A^j}$ [34]. The interarrival time, $v_{A^j}$, with parameter $\lambda_{A^j}$ has the probability distribution function:

$$P\left(v_{A^j}\right) = \lambda_{A^j} \cdot e^{-\lambda_{A^j} \cdot v_{A^j}} \tag{2}$$

The expectation of the number of packets fed to the input of tunnel $j$ is:

$$\mathbb{E}\left(v_{A^j}\right) = \lambda_{A^j} \tag{3}$$

It is considered that the times necessary to transmit the packets, i.e., the service times on different tunnels, are independent and identically distributed with probability distribution function $F_{B_j}(\cdot)$ and probability density $f_{B_j}(\cdot)$. To ensure the stability of the queue the users' service flows must be combined, i.e., distributed on the available tunnels, so that the occupation rate $\rho_j$ is less than one [34] on each tunnel.

$$\rho_j = \frac{\lambda_{A^j}}{\mathbb{E}\left(B_j\right)^{-1}} < 1 \tag{4}$$

where $\mathbb{E}\left(B_j\right)^{-1}$ represents the expectation of the packet rate on tunnel $j$.

The state of the $j$-th tunnel can be described at a time moment by the tuple $(n_j, \tau_j)$ where $n_j$ is the number of packets stored in the queue of the tunnel in discussion and $\tau_j$ is the delay experienced by a packet in the transmission chain. The first parameter of the tunnel's state is discrete, while the second one is continuous, and this could complicate the analysis. However, if we consider only the queuing process of the tunnel, then the tunnel's state description can be simplified to $n_j$ only, $\tau_j$ being 0 for the new packets stored in the queue. We denote by $L^d_{j,k}$ the number of packets remaining in the queue after the

transmission of the *k*-th packet, and by $d_{n^j}$ the fraction of packets that leave in the queue $n_j$ packets after their transmission [34].

$$d_{n_j} = \lim_{k \to \infty} P\left(L_{j,k}^d = n_j\right) \tag{5}$$

The number of packets remaining in the queue after the *k*+1-th packet is sent, $L_{j,k+1}^d$, is equal to the number of packets present when the *k*-th packet was sent minus one plus the number of packets that are stored in the queue during the transmission time of the *k*+1-th packet, i.e., during the service time of packet *k*+1. This last number of packets is denoted by $A_{j,k+1}$ and we have [34]:

$$L_{j,k+1}^d = L_{j,k}^d - 1 + A_{j,k+1} \; ; if \; L_{j,k}^d > 0$$
$$L_{j,k+1}^d = A_{j,k+1} \; ; if \; L_{j,k}^d = 0 \tag{6}$$

From the equations above result that the sequence $\left\{L_{j,k}^d\right\}_{k=0}^{\infty}$ can be represented as a Markov chain, the transition probabilities between the states of this chain being given by:

$$p_{i,l} = P\left(L_{j,k+1}^d = l \mid L_{j,k}^d = i\right) \tag{7}$$

By $\alpha_{n_j}$ we denote the probability that during the transmission of a packet exactly $n_j$ packets arrive. Considering that during the service time, *t*, of a packet the number of new packets that are stored in the queue is Poisson distributed with parameter $\lambda_{A^j} \cdot t$ we can express $\alpha_{n_j}$ as:

$$\alpha_{n_j} = \int_{t=0}^{\infty} \frac{\left(\lambda_{A^j} t\right)^{n_j}}{n_j!} e^{-\lambda_{A^j} \cdot t} f_{B_j}\left(t\right) dt \tag{8}$$

The limiting probability $d_{n_j}$, defined according to (5), satisfies the equilibrium equations of the Markov chain attached to the queuing process [34]:

$$d_{n_j} = d_{n_j+1}\alpha_0 + d_{n_j}\alpha_1 + \cdots + d_1\alpha_{n_j} + d_0\alpha_{n_j} =$$
$$= \sum_{k=0}^{n_j} d_{n_j+1-k}\alpha_k + d_0\alpha_{n_j}, \quad n_j = 0,1,\dots \tag{9}$$

If the queue is in equilibrium the maximum value of $n_j$ will be limited because the fraction of the packets leaving behind in the queue $n_j$ packets, after their transmission, is decreasing with the value of $n_j$ and will approach zero for $n_j$ large.

We denote the total time spent in the system by a packet with the random variable *S* having a probability distribution function $F_{S_j}\left(\cdot\right)$ and probability density function $f_{S_j}\left(\cdot\right)$. It is considered that the system is in equilibrium when a new packet arrives in the queue. The distribution of the number of packets stored in the buffer after the transmission of this (new) packet is equal to $\{d_{n_j}\}_{n_j=0}^{\infty}$. Considering a FIFO system we have, similar to Equation (8):

$$d_{n_j} = \int_{t=0}^{\infty} \frac{\left(\lambda_{A^j} t\right)^{n_j}}{n_j!} e^{-\lambda_{A^j} \cdot t} f_{S_j}\left(t\right) dt \tag{10}$$

The time spent in the system by a packet (also called sojourn time), is the sum of *W* (its waiting time in the queue) and *B* (its transmission time on the channel), where *W* and *B* are independent random variables. More precisely a newly arriving packet in the queue first must wait for the service time of the packet that is in the transmission chain (residual service time), and then it continues to wait for the transmission of all packets which were already in the queue on its arrival. Based on PASTA (Poisson Arrivals See Time Averages) the server of the queue is busy on the arrival of a new packet with probability $\rho_j$, which in our case is the occupation rate of the tunnel. We denote by the random variable $R_j$ the residual service time, and according to [34] we have the following relation between the waiting time in the queue, service time (transmission time in the tunnel), residual service time, and the number of packets waiting in the queue:

$$\mathbb{E}\left(W_j\right) = \mathbb{E}\left(n_j\right)\mathbb{E}\left(B_j\right) + \rho_j\mathbb{E}\left(R_j\right) \tag{11}$$

and according to Little's law, we have:

$$\mathbb{E}\left(n_j\right) = \lambda_{A^j}\mathbb{E}\left(W_j\right) \tag{12}$$

and finally, we obtain:

$$\mathbb{E}\left(W_j\right) = \frac{\rho_j\mathbb{E}\left(R_j\right)}{1-\rho_j} \tag{13}$$

The value of the mean residual service time may be written in the form [34]:

$$\mathbb{E}\left(R_j\right) = \frac{\mathbb{E}\left(B_j^2\right)}{2\mathbb{E}\left(B_j\right)} = \frac{\sigma_{B_j}^2 + \mathbb{E}\left(B_j\right)^2}{2\mathbb{E}\left(B_j\right)} = \frac{1}{2}\left(c_B^2+1\right)\mathbb{E}\left(B_j\right) \tag{14}$$

An important observation resulting from (13) and (14) is that the mean waiting time only depends on the mean and standard deviation of the transmission time random variable and not on the distribution function of this variable. It results that, in practice, it is enough to compute the mean and standard deviation of the service time to estimate the mean waiting time. Based on the previous relations the mean value of the system's response time (the so-called sojourn time) can be expressed as:

$$\mathbb{E}\left(S_j\right) = \frac{\rho_j}{1-\rho_j}\frac{1}{2}\left(c_B^2+1\right)\mathbb{E}\left(B_j\right) + \mathbb{E}\left(B_j\right) \tag{15}$$

If the service time has a finite expectation $\mathbb{E}\left(B_j\right) = \mu_j^{-1}$ and a finite standard deviation, $\sigma_{B_j} = \sigma_j^{-1}$ then the expected sojourn time with Poisson arrivals with rate $\lambda_j$ (obeying condition (4)) is the following:

$$\mathbb{E}\left(S_j\right) = \frac{\lambda_j\left(1+\sigma_j^2\mu_j^2\right)}{2\mu_j\left(\mu_j-\lambda_j\right)} + \frac{1}{\mu_j} \tag{16}$$

### 2.1.2. User Clustering in Multi-AP Scenarios

In some circumstances, there could be available several small cells to which the users can connect. For example, several small cells could be deployed in a building, or several APs could be installed in a train to which the users can connect. In this case, the users' access also should be controlled, i.e., should be identified the user groups/clusters that will connect to each AP. This problem involves several theoretical and practical issues:

- the user-AP link should fulfill some QoS requirements, and the simplest solution is to connect the user terminal to the closest AP or to the AP with the largest transmission power;
- the users should connect with higher priority to the AP having more available transmission resources on the communications tunnels established over the cellular wireless links. This is necessary to avoid the overloading of some APs, with effect over the global QoS, and for efficient usage of the resources available on the tunnels;
- the selection process of the users who connect to one of the APs should avoid the need to install supplementary software modules on the user terminals, which would complicate the control of the users' access process.

The transmission scenario envisioned is depicted in Figure 4a. All the users are in the coverage area of AP-A and if no other AP is installed all users will connect to this AP. A second AP, AP-B, is installed, the users also being in the coverage area of this AP. The access protocol should select the users which connect to each AP, the selection being a dynamic one based on the transmission power of the APs and the available transmission resources on the communication tunnels used as backhaul connections. The available resources depend on the virtual tunnels' instantaneous capacities and the instantaneous packet and bit rates of the users' data flows. This scenario could be extended to more complex ones with more APs. For example, the train deployment scenario could include a third AP, AP-C (see Figure 4b), with only some of the users being located in the overlapping coverage areas, i.e., the AP-A: AP-B respectively the AP-A : AP-C coverage area, while all the users are located in the coverage area of AP-A. The building deployment scenario could include more APs, installed on the same or ifferent floors of the building, and the users located in the overlapping coverage areas of several APs.

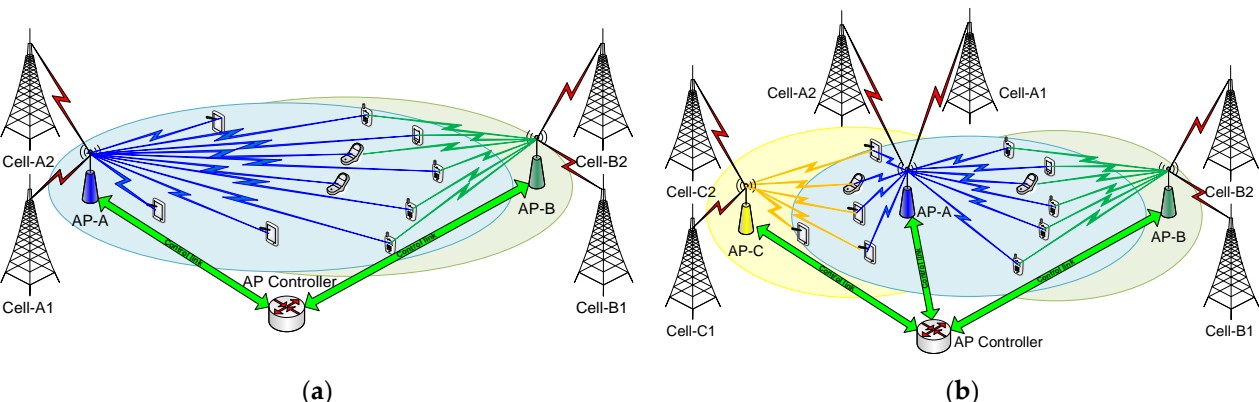

(**a**)             (**b**)

**Figure 4.** Networking scenario with several APs deployment: (**a**) 2 AP deployment; (**b**) 3 AP deployment.

The users' clustering algorithm could be a decentralized one, when no central control unit is employed, or could be a centralized one when an AP Controller module is used. The presence of this module does not raise significant technical issues, the APs being part of an infrastructure network, of a building, or of a vehicle, that could integrate such a control module. The AP Controller module also could be one of the APs acting as a master.

It should be noted that, if several APs are used the available communication tunnels will be divided between the APs, each AP selecting several users that can be served with the resources available on the allocated virtual tunnels. The number of communication tunnels is fixed, the tunnels being leased from the macro cell operators by the train or building operator, and this fixed number of tunnels should be divided between the installed APs, the users being also split between the installed APs.

*2.2. Modeling the Load Balancing on a Set of Tunnels as a Multiplayer Game*

We consider that our communication system must transmit $K$ input flows and there are available $N$ tunnels. We denote by $\mathbf{s} = \left( \mathbf{K_1}, \mathbf{K_2}, ..., \mathbf{K_N} \right)$ the surjective mapping of the $K$ flows on the tunnels. $\mathbf{K_j}$ denotes the set of input flows mapped to tunnel $j$, and we have the conditions: $\mathbf{K_1} \cup \mathbf{K_2} \cup ... \cup \mathbf{K_N} = \mathbf{K}$ and $\mathbf{K_i} \cap \mathbf{K_j} = \varnothing; \forall i \neq j$. We denote by $\mathbf{S}$ the set of all flow-tunnel mapping possibilities and the system response function $r : \mathbf{S} \rightarrow \mathbb{R}; \mathbf{s} \in \mathbf{S}$ gives the system's efficiency if mapping $\mathbf{s}$ is used. The objective of the LB algorithm is to identify a mapping that maximizes the system efficiency [35], which can be defined as:

$$arg_s \max r(\mathbf{s}) := \left\{ \mathbf{s} \mid \forall x \in \mathbf{S} : r(x) \leq r(\mathbf{s}) \right\} \tag{17}$$

The LB problem defined above can be modeled as a game in which players are agents representing the flows and the tunnels [36]. The strategies of the flow agents consist of the selection of the tunnels established through the heterogeneous network. The objectives of the flow agents are the maximization of the flows' throughput which is equivalent with the minimization of the delay suffered in the system by the packets of the flows (private objective). The strategies of the tunnel agents consist of the selection of a mapping $\mathbf{s}$, so that the total rate of the flows mapped on a tunnel should be equal to or less than the instantaneous capacity $C_i(t)$ of the tunnel represented by the agent.

### 2.2.1. Selfish Routing Load Balancing

In this game, each flow agent chooses the route (tunnel) which minimizes the transit time of the flows' packets (private objective of the game). The social goal of the game is to find an arrangement which minimizes the average delay. Following [37], the notion of "selfishly defined traffic flow" is formalized in the next theorem. In this case, it is expected that each packet of a flow is transmitted along the path inserting the minimum latency, the latency being measured with respect to the rest of the flow's packets. If this condition can't be fulfilled for some packets, those packets will be routed on the path with the smallest possible delay. The latency $\ell_{T_j}(f_i)$ introduced by the tunnel $T_j$ on the packets of flow $f_i$, i.e., $f_i \in \mathbf{K}_j$, is defined by the sojourn time.

**Theorem 1.** *A flow $f_i$ in a graph G representing the network is at Nash equilibrium if and only if for every $i \in \{1, ..., K\}$ $f_i \in \mathbf{K}_j$ $\ell_{T_j}(f_i) \leq \ell_{T_z}(f_i); \forall z \in \{1, ..., N\}, z \neq j$.*

Briefly, Theorem 1 state that, in a flow at Nash equilibrium, all packets travel on the minimum latency paths. If $f_i$ is at Nash equilibrium, then all flows mapped to that tunnel have equal latency let say $L_j(\mathbf{K}_j)$. Therefore, the social cost $\mathcal{C}(\mathbf{s})$ of an arrangement can be expressed as [36]:

$$\mathcal{C}(\mathbf{s}) = \sum_{j=1}^{N} L_j(\mathbf{K}_j) \cdot \lambda_j \tag{18}$$

where $\lambda_j$ denotes the average packet rate through tunnel $j$.

If the stability condition is fulfilled (i.e., the global input rate is less than or equal to the total capacity of the tunnels) the Nash equilibrium always exists and is inherently unique.

Definition of the game:

- the $f_i, i \in \{1, ..., K\}$ flow-agents, i.e., the players;
- $A_i = \{T_1, ..., T_N\}$ is the finite set of the actions of player $i$, i.e., each player (flow) should choose one tunnel. It is defined an action profile $\mathbf{a} = (a_1, a_2, ..., a_K)$, where each element

$a_i$ of the vector represents one action for each player of the game, and this profile corresponds to an arrangement **s**;

- the payoff or the utility function $u_i$ of a player (flow) in this case is $u_i = \ell_{T_j}(f_i)$, i.e., the expected latency if the agent $i$ chooses tunnel $j$.

Algorithm 1 presented below gives the operations involved by the Selfish Routing LB algorithm.

---

**Algorithm 1** Selfish Routing-based LB algorithm

---

Compute the expected latency for each packet of the flow $f_i$ on every tunnel.

1: **for** $j = 1$ to **N do**

2:     compute $\ell_{T_j}(f_i)$ based on (16).

3: **end for**

Select for flow $f_i$ the tunnel which ensures the minimum latency.

4: $z = \arg\min_{j=1,...,N}\left(\ell_{T_j}(f_i)\right)$

Associate flow $f_i$ to tunnel $z$.

5: $\mathbf{K}_z = \mathbf{K}_z \cup \{f_i\}$

6: Generate the signaling traffic for flow routing and send it on default tunnel $j = 1$

---

In the previous sections it was mentioned that the mixture of flows applied to each tunnel is supposed to be Poisson distributed, and in consequence the packets interarrival times are exponentially distributed with parameter $\lambda_{A^j}$. This parameter is necessary to estimate the expected latency $\ell_{T_j}(f_i)$ on each tunnel. The $\lambda_{A^j}$ parameter can be obtained by fitting the measured distribution of the interpacket delays with the pdf curve of the exponential distribution. By using the method of moments results $1/\lambda_{A^j} = mean(\Delta p_j)$, where $\Delta p_j$ are the packets interarrival times of the flow mix applied to tunnel $j$. In this simple case, we have only one parameter of the distribution which should be computed.

Even if this algorithm has low complexity, $\mathcal{O}(N)$, the social cost of the Nash equilibrium (NE) is worse than the optimal solution, i.e., due to the lack of coordination among the players the solution obtained at Nash Equilibrium is not the optimal one, and the cost of this lack of coordination between players is called the *price of anarchy* and is defined as [37]:

$$\max_{NE} \frac{C(NE)}{C(opt)} = \text{price of anarchy} \tag{19}$$

Using a set of $N$ tunnels with an expected total transfer rate at least $\mu_{min}$, the price of anarchy for situations when the latency functions are defined by (16) and the sum of all flow rates transmitted over each tunnel is at most $R_{max} \leq \mu_{min}$, is given by [38]:

$$\max_{NE} = \sup_{j\in\{1,...,N\}}\left(1+\sqrt{\frac{\mu_j}{\mu_j - R_{max}}}\right) \frac{2\mu_j + R_{max}\left(\sigma_j^2\mu_j^2 - 1\right)}{4\mu_j + \left(\mu_j + R_{max} - \sqrt{\mu_j\left(\mu_j - R_{max}\right)}\right)\left(\sigma_j^2\mu_j^2 - 1\right)} \tag{20}$$

where $\mu_j$ and $\sigma_j$ denotes the expectation respectively the standard deviation of the packet transfer rate distribution associated with tunnel $j$.

The algorithm involves a high signaling overhead required to change the tunnel associated with the flow, change of tunnel which could happen with each new packet arriving at the AP. To reduce the amount of computation and signaling required by the algorithm a timer can be defined for each flow, and the algorithm is executed only when the

timer expires and not when a new packet arrives at the AP node. The LB solution remains unchanged until the timer expires, i.e., during the timer period. Changing the tunnel at each packet could generate a large amount of jitter which might not be compensated by the buffering operations performed at the receiver. The amount of jitter generated will be reduced if the algorithm is executed only when a timer expires.

2.2.2. Auction-Based Load Balancing

To describe the LB process as a *first price sealed-bid auction* [39], the LB module could be considered a contractor who has $K = |\mathbf{K}|$ goods (the $K$ flows) to deliver and let the $N$ tunnel agents represent the $N$ available deliverers. Each tunnel agent $j$ submits simultaneous "sealed bids" $b_j(f_i)$ which represent the price for transferring the packets of flow $f_i$, the bidder with the lowest offer wins the contract for transferring flow $f_i$, and the LB module pays the price for the transfer of the flow $f_i$. The goal of the tunnel agents is to maximize their payoff represented by the difference between the price paid by the LB module, $b_j(f_i)$, and the true cost of transferring the flow $f_i$ through the tunnel $j$, denoted by $v_j(f_i)$. If $b_j(f_i)$ is not the winning bid, then the payoff to tunnel agent $j$ is 0. If $b_j(f_i)$ is the winning bid, then the payoff to tunnel agent $j$ is $b_j(f_i) - v_j(f_i)$. It can be noticed that the true-value bidding in this game cannot be the dominant strategy. By bidding for the true value, a bidder receives a payoff of 0 if it loses and it will also receive a payoff of 0 if it wins, since he receives the same value as its costs. As a result, the optimal way to bid in the first price auction is to "shade" the bid slightly upward, so that if a bidder wins, it receives a positive payoff. Finding out how much to increase the bid involves a trade-off between two opposite forces. If the bid is close to the true value, then the payoff will be small if the bidder wins. However, if the asked price is too large compared to the true costs, this increases the potential payoff of the bidder in the event of winning, but it reduces the chance of being the lowest bid and in consequence the chance of winning. Finding the optimal value of the bid is a complex problem that depends on the knowledge of the other bidders and their way of acting.

In the case of our optimization problem, we consider that the real cost of carrying the flow $f_i$ through the tunnel $j$ would be:

$$v_j(f_i) = L_j\left(\mathbf{K}_j \cup \{f_i\}\right) \cdot \left(\lambda_j + \lambda_i\right) \tag{21}$$

which represents the expected latency (relative to the interpacket delay) inserted by tunnel $j$ if it wins the delivery contract of flow $i$. The submitted bid is proportional to the communication resources, i.e., the bandwidth used to ensure the requested delays when the packet rates have some values, and the used bandwidth can be considered the cost necessary to carry the packets.

For simplicity it can be considered that the submitted bid by the tunnel is:

$$b_j(f_i) = \left\{ L_j\left(\mathbf{K}_j \cup \{f_i\}\right) \cdot \left(\lambda_j + \lambda_i\right) + n_j \right\} + \Delta_{i,j} = \left\{ v_j(f_i) + n_j \right\} + \Delta_{i,j} \tag{22}$$

It can be noticed that the bid submitted by the agent $j$, is shaded by the number of packets waiting in the queue $n_j$, while $\Delta_{i,j}$ represents the amount of penalty which the contractor (the LB module) must pay if it changes the beneficiary of a contract:

$$\Delta_{i,j} = \begin{cases} n_{z \to j} & ; f_i \notin \mathbf{K}_j \\ 0 & ; f_i \in \mathbf{K}_j \end{cases} \tag{23}$$

where $\underset{z \to j}{n_i}$ represents the amount of signaling traffic (number of signaling packets) required to move the flow *i* from the tunnel *z* to tunnel *j*.

The number of packets waiting in the queue $n_j$ will influence the evolution of the delay in the next time periods, and to keep the desired value of the delay it is possible to be needed more bandwidth if $n_j$ increases, i.e., the cost for carrying the packets is influenced by $n_j$.

Algorithm 2 presented below gives the operations involved by the Auction-Based LB mechanism.

---

**Algorithm 2** Auction-based load-balancing algorithm

Each new traffic initially is associated with default tunnel *j* = 1.

At each arriving packet

1: **for** *j* = 1 to **N do**

2:       estimate $v_j\left(f_i\right)$ based on (21).

3:       compute $b_j\left(f_i\right)$ based on (22) and (23).

4: **end for**

Select for flow $f_i$ the tunnel which submitted the lowest bid.

5: $z = \arg \min_{j=1,\dots,N}\left(b_j\left(f_i\right)\right)$

Associate flow *i* to tunnel *z*

6: $\mathbf{K}_z = \mathbf{K}_z \cup \left\{f_i\right\}$

Update the tunnel total revenue after sending the packet.

---

The difference between this algorithm and the previous one is that here all packets belonging to one flow are carried by the winning tunnel. A flow remains attached to a tunnel until its global quality degrades so much that it is more beneficial to that flow to change the tunnel (even "paying" the penalty). The algorithm's complexity is $\mathcal{O}\left(N\right)$ for each arriving packet. As in the situation of the Selfish Routing LB algorithm, the amount of computation required can be decreased by defining some timers attached to each flow and executing the algorithm only when the timer expires and not for each new packet arriving at the AP. The downside of such a solution is the slower response of the system to the changes in the input parameters.

2.2.3. Combinatorial Auction-Based Load Balancing

It is considered that the contractor from the previous algorithm could receive quotes for one single flow at a time. However, this may not be very effective because the auction should be repeated with each arriving packet. The combinatorial auction allows players to bid on a subset $\mathbf{K}_z \subseteq \mathbf{K}$. These are most useful if there are *compliments*, i.e., a set of items might be worth more than the sum of the parts [37]. In this case, the bidders submit bids $b_j\left(\mathbf{K}_z\right)$ on each $\mathbf{K}_z \subseteq \mathbf{K}$. The auctioneer chooses an allocation $\mathbf{s} = \left(\mathbf{K}_1^*, \mathbf{K}_2^*, \dots, \mathbf{K}_N^*\right)$ that minimizes $\sum_{j=1}^{N} b_j\left(\mathbf{K}_j\right)$ over all feasible allocations. Considering all possible arrangements each player submits an exponential number of bids, and the allocation problem becomes NP-hard [36].

To reduce the dimension of the optimization space, the flow agents are endorsed with a potential function $\phi_i$. These potential functions describe the time evolution of the beneficiary's satisfaction with the deliverer [40], i.e., the variation in time of the key performance indicators (bit rate and latency) of the flows. If the potential functions of the flows mapped on a tunnel decrease, the contractor decides to revise the contracts for that tunnel.

It generates a set of allocations redistributing a subset of the flows mapped on that tunnel. Considering that the tunnel agents are single-minded bidders, i.e., player *j* only cares about one subset **K**$_j$, the computation complexity is reduced to polynomial time.

In the case of delay-sensitive flows, the time spent by a packet in the transmission chain is the primary parameter that influences the QoS of the transmission. In this case, the *potential function* should express the delays experienced by the packets of the flow and the degree of user satisfaction in the condition of the experienced delays, which decreases rapidly as the delay is increasing. Denoting by $\tau_i$ the maximum value of the expected delay, we define the utility function for the delay-sensitive data streams as:

$$\phi_i = 10^{-\frac{\tau_i}{c_i}} \tag{24}$$

where $c_i$ is a constant value associated with each flow which allows differentiating the same type of real-time service flows. The maximum value of the proposed utility function is one. Figure 5a presents the variation of the utility function given in (24) with the delay inserted by the tunnel when the coefficient $c_i$ has different values. The used potential function allows a good separation between real-time flows with different maximum delay constraints.

For non-real-time traffic the main parameter of interest is the average throughput. The experienced delay, and the possible variations of this delay, have significantly lower importance, even if the delay inserted by the transmission chain cannot be neglected. The proposed potential function expresses the satisfaction of the user according to the Average Call Throughput associated with the flow and the target is, of course, the maximization of the user's satisfaction. The Average Call Throughput $R_i$ can be defined as the total number of transmitted bits divided by the duration of the transmission process. Let $R_i^{call}$ denote the number of bits sent by the user generating flow *i* during the current call and $t_i^{call}$ the time elapsed from the beginning of the current call. If the number of bits that can be transferred through the tunnel *j* in a single time unit (for ex. one millisecond) is $L_j$, the instantaneous value of the average call throughput, computed on a finite length window, will be:

$$R_i = \frac{R_i^{call} + L_j}{t_i^{call}} \tag{25}$$

The $L_j$ parameter results from the instantaneous capacity of the wireless link $C_j(t)$.

We define the potential function associated with the delay tolerant traffic as:

$$\phi_i = \tanh\left(\frac{R_i}{R_i^{av}}\right) = \frac{e^{\frac{2R_i}{R_i^{av}}} - 1}{e^{\frac{2R_i}{R_i^{av}}} + 1} \tag{26}$$

where $R_i^{av}$ is the expected value of the average bit rate and can be computed according to (1). This parameter allows the differentiation between different non-real-time flows. The maximum value of this potential function is also 1. The potential function was defined in such a way as to express user satisfaction with the increasing values of the throughput. Figure 5b shows the variation of the proposed utility function with the instantaneous bit rate, $L_j$, when the expected average bit rate $R_i^{av}$ has different values. The hyperbolic tangent function is a good choice for representing the progressive transition between two extreme states in our situation: not satisfied at all and totally satisfied with the throughput provided by the system. It also allows a good separation between non-real-time services with different average rate constraints. The operations involved by the Combinatorial Auction-Based LB mechanism are described in Algorithm 3, presented below.

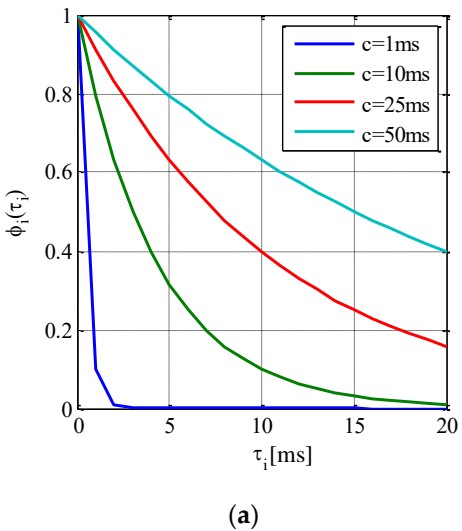
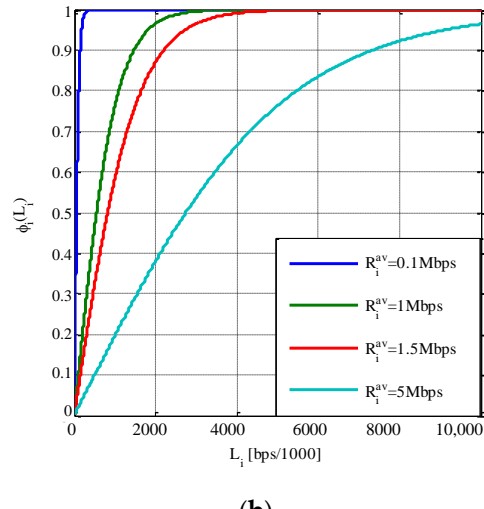

(**a**)                                                    (**b**)

**Figure 5.** Potential functions for delay sensitive and best effort traffic types: (**a**) Variation of the potential function for delay sensitive traffic with latency $\tau_i$ when coefficient $c_i$ has different values; (**b**) Variation of the potential function for best effort traffic with the instantaneous bit rate of the tunnel, $L_i$, for different values of the expected average bit rate $R_i^{av}$.

---

**Algorithm 3** Combinatorial Auction-based load-balancing algorithm

Each new traffic initially is associated with default tunnel $j = 1$.

At each arriving packet update the flows' potential function $\phi_{f_i}$.

In each moment $mT$; $m = 0, 1, 2,\dots$; $T$–interpacket delay or timer period, perform the following steps:

1: **for** $j = 1$ to **N do**

2:       compute $\Phi_j^m = \dfrac{\sum\limits_{f_i \in \mathbf{K}_j} \phi_{f_i}}{\left|\mathbf{K}_j\right|}$

3:       compute $\Delta\Phi_j^m = \Phi_j^m - \Phi_j^{m-1}$

4:       **while** $\Delta\Phi_j^m < imposed\ treshold$ **do** // the QoS decreases considerably

Select the flow with the lowest potential.

5:       $i = \arg\min\limits_{f_i \in \mathbf{K}_j}\left(\phi_{f_i}\right)$

Create a new arrangement:

6:       $\mathbf{s} = \left(\mathbf{K_1} \cup \left\{f_i\right\}, \mathbf{K_2} \cup \left\{f_i\right\},\dots,\mathbf{K}_j \cap \left\{f_i\right\},\dots,\mathbf{K_N} \cup \left\{f_i\right\}\right)$

7:           each bidder $z \neq j$ submits a bid $b_z\left(\mathbf{K}_z\right)$ for the new subset $\mathbf{K}_z$

Route flow $f_i$ through the tunnel which submitted the lowest bid.

8:       $t = \arg\min\limits_{\substack{z=1,\dots,N \\ z \neq j}}\left(b_z\left(\mathbf{K}_z\right)\right)$

Associate flow $i$ to tunnel $t$.

9:       $\mathbf{K}_t = \mathbf{K}_t \cup \left\{f_i\right\}$

10:       recompute $\Phi_j^m = \dfrac{\sum\limits_{f_i \in \mathbf{K}_j} \phi_{f_i}}{\left|\mathbf{K}_j\right|}$

11:       recompute $\Delta\Phi_j^m = \Phi_j^m - \Phi_j^{m-1}$

12:       **end while**

13: **end for**

*2.3. Reference Load-Balancing Algorithms*

To assess the performances of the proposed GT-based LB algorithms two simpler and less complex algorithms were considered as references.

2.3.1. The Round Robin Reference Load-Balancing Algorithm

This algorithm represents the simplest scheduling algorithm used in packet switched networks. In this case, it was converted into a LB algorithm over parallel tunnels as it is described below by Algorithm 4.

---
**Algorithm 4** Round Robin Load-Balancing Algorithm

Each new data flow initially is associated with default tunnel $j = 1$
1: **if** $f_i$ is a new flow
2: identify the last used tunnel $z \leq N$
3: select the next tunnel: $z = (z+1) \bmod N$ and send flow $f_i$ on tunnel $z$
4: **end if**

---

This algorithm does not consider either the capacity of the tunnels or the characteristics of the flows, i.e., type of flow, bit rate, etc. The transfer rate on a given tunnel necessary to transmit all the packets routed on that tunnel could be significantly larger than the capacity of the tunnel, which will have consequently low performance.

2.3.2. The Multiple Knapsack Reference Load-Balancing Algorithm

The Multiple Knapsack algorithm tries to distribute a set of items on a set of containers, called knapsacks. Each knapsack is characterized by its capacity, and the knapsacks represent the tunnels in our scenario. Each item is characterized by a weight parameter and a value or utility. The items are representing in our scenario the flows, characterized by an average rate (weight) and a priority (utility). The algorithm tries to distribute the items on the knapsacks in such a way as to maximize the total utility while fulfilling the capacity limitations of the knapsacks [41,42].

The variant of this algorithm which was implemented and simulated is described by Algorithm 5 blows

---
**Algorithm 5** Multiple Knapsack load-balancing algorithm

Each new data flow initially is associated with default tunnel $j = 1$.
1: **do**
2: compute the priority of the new data flows.
3: sort the flows in descending order of their priorities.
4:     **if** flows $f_1, f_2, \ldots f_k$ have the same priority
5:         sort flows $f_1, f_2, \ldots f_k$ in ascending order of their average bit rate.
6:     **end if**
7: route the sorted flows on the available tunnels while fulfilling the conditions:
      a. the sum rate of the flows routed on a tunnel is lower than the average capacity of the
      tunnel **and**
      b. the sum of the priority of the flows routed through a tunnel is as high as possible.
8:   **while** each flow is associated with a tunnel, **or** no tunnels are available
9:       **if** flow $f_i$ does not fit on one of the tunnels
10:         check if it fits into one of the other available tunnels.
11:       **end if**
12:       **if** flow $f_i$ cannot be routed over either of the tunnels
13:         flow $f_i$ will be routed over the default tunnel or will be rejected.
14:       **end if**
15:   **end while**
14: **end do**

---

In what concerns the priorities of the flows, five classes were considered: class 1-gaming, video conferences, class 2-video streaming, class 3-voice over IP, class 4-web browsing using HTTP, and class 5-best effort FTP. In a simpler approach, the flows can be classified into only two classes, i.e., real-time with high priority and non-real-time with low priority. This algorithm works at flow level (not at packet level), and it is executed when a new flow is generated, or when a predefined timer expires, or the parameters of the networks (e.g., the quality of the wireless links) change significantly.

*2.4. Modeling the User Clustering Process as a Game*

The user clustering problem aims to associate each user terminal to one and only one AP in such a way as to maximize the use of the available resources while fulfilling the flows' end-to-end QoS requirement. For this, each user should be connected to the nearest AP (to maximize the user-AP link's quality), and the difference between the load and capacity of the APs should be minimized. To model the clustering problem, let's denote by $D_i(t)$ the demand formulated by the user *i*, and by $C_j(t)$ the sum of the capacity of the tunnels starting from APs *j*. Let $x_{i,j}$ be a binary variable which is $x_{i,j} = 1$ if the user *i* is connected to APs *j* and 0 otherwise, then the load $L_j(t)$ of the *j*-th AP is:

$$L_j(t) = \sum_i x_{i,j} \cdot D_i(t) \; \forall j \tag{27}$$

The clustering problem can be formulated as a liner programming problem in the following way:

$$\begin{aligned} &\min_j \sum_j y_j \cdot \left(L_j(t) - C_j(t)\right) \\ &\min_j \sum_i \sum_j x_{i,j} \cdot d_{i,j} \\ &\quad s.t. \\ &\sum_j x_{i,j} = 0 \; \forall i \\ &x_{i,j}, y_j \in \{0,1\} \end{aligned} \tag{28}$$

where $y_j = 1$ if $L_j(t) > C_j(t)$ and 0 otherwise and $d_{i,j}$ is the distance between the *i*-th user terminal and the *j*-th AP.

To solve the user clustering problem a simple non-cooperative and a cooperative game is proposed. In each case, the AP is represented by a "user broker" (player) $ap_i$, which allows more users to connect to the AP if the AP's load is less than the instantaneous capacity. In the non-cooperative game, each AP agent chooses the action that minimizes the gap between the load and backhaul capacity of the represented AP.

Definition of the game:

- the $ap_i$, $i \in \{1,\dots,K\}$ AP agents are the players;
- $A_i = \{P_1,\dots,P_N\}$ is the finite set of the actions of player *i*, i.e., each player should choose a transmit power that influences the user terminals to connect or not to the specific AP. It is defined an action profile **a** = ($a_1$, $a_2$, …,$a_K$), where each element $a_i$ of the vector represents one action for each player of the game, and each action profile corresponds to an arrangement **s** of APs and users.

- the payoff or the utility function $u_i$ of a player in this case is $u_i = C_i(t) - L_i(t)$, i.e., the expected difference between the load and the capacity of $AP_i$.

The steps necessary to implement the non cooperative clustering process are described by Algorithm 6.

| **Algorithm 6** Non cooperative user clustering mechanism |
|---|
| estimate the difference between the demand and capacity for each AP. |

1: **for** *j* = 1 to N **do**

2:      estimate $C_i(t) - L_i(t)$ using the average packet delay or the lengths of the transmit queues.

Check if the AP has free capacity.

3:      **if** $C_i(t) - L_i(t) > 0$

Increase the transmit power of APi and by this increase the coverage area and the number of users connected to APi.

4:      **else**

Reduce the transmit power of APi and by this reduce the coverage area and the number of users connected to APi.

5:      **end if**

6: **end for**

7: Each user checks the strengths of the received signal from each AP and chooses the AP from which the received signal has the highest power.

In the case of the cooperative game each AP agent, *api*, collaborates with the agents of the neighboring APs to choose the strategy that might minimize the gap between their load and the backhaul capacity.

Definition of the game:

- the $ap_i$, $i \in \{1,…,K\}$ AP agents are the players;

- $A_{i,j} = \{(P_1,P_1),(P_1,P_2),…,(P_N,P_N)\}$ is the finite set of the actions of the tuple of players $(ap_i, ap_j)$, i.e., each group of neighbor player should adjust their transmit power that influences the user terminals to connect or not to the specific AP. An action profile **a** = $(a_1, a_2, …,a_K)$ is defined, where each element $a_i$ of the vector represents one action for each tuple of players, and each action profile corresponds to an arrangement **s** of APs and users;

- the payoff or the utility function $u_{i,j}$ of a group of players in this case is $u_{i,j} = |L_i(t) - C_i(t)| + [|L_j(t) - C_j(t)|]$, i.e., the expected difference between the total load and the sum of APs' capacity.

The steps necessary to implement the cooperative clustering process are described by Algorithm 7.

---

**Algorithm 7** Cooperative user clustering mechanism

---

estimate the difference between the demand and capacity for each group of two neighbor Aps.

1: **for** i = 1 to N **do**

2:      **for** j = i+1 to N **do**

3:      estimate $\left|\dfrac{L_i(t) - C_i(t)}{C_i(t)}\right|$ and $\left|\dfrac{L_j(t) - C_j(t)}{C_j(t)}\right|$ using the average packet delay or the lengths of the transmit queues.

Check if the APs are load balanced.

4:      **if** $\left|\dfrac{L_i(t) - C_i(t)}{C_i(t)}\right| > \left|\dfrac{L_j(t) - C_j(t)}{C_j(t)}\right|$

Increase the transmit power $P_i$ of APi and decrease the transmit power $P_j$ of APj.

5:      **else**

Decrease the transmit power $P_i$ of APi and increase the transmit power $P_j$ of APj.

6:      **end if**

7:      **end for** j

8: **end for** i

9: Each user checks the strengths of the received signal from each AP and chooses the AP from which the received signal has the highest power.

---

## 3. Results

*3.1. The Architecture of the Simulation Platform Used for the Evaluation of the LB Algorithms*

The architecture of the simulation platform used to assess the performance of the proposed LB algorithms is depicted in Figure 6.

The Traffic Generator module generates the data flows for each user according to the specified traffic mixture and statistical characterization of each traffic type [43]. The generated data flows are routed through the tunnels based on the decisions taken by the LB module. The Wireless Channel Simulator module simulates the wireless links instantiated in the cellular networks according to the considered channel models. To each simulated wireless link a Channel Estimator module is associated, these modules perform a short-term prediction of the channel parameters based on the latest simulated/computed values. In this way, it is possible to make a short-term prediction of the tunnels' instantaneous capacity. The LB module receives the statistical data characterizing the generated data flows, the Queue State Information (QSI) associated with each tunnel, and the Channel State Information (CSI) generated by the Wireless Channel Simulator module. Using all this information, for each input flow, the LB module tries to choose the tunnel which ensures the smallest delay accumulated through the system. At the output of the tunnels, the accumulated delay and other performance indicators are evaluated for each packet, and statistical data are collected for each flow. These results together with the data describing the status of the system can be stored in a database for further evaluation. The following subsections describe each constituent block and the considered algorithms.

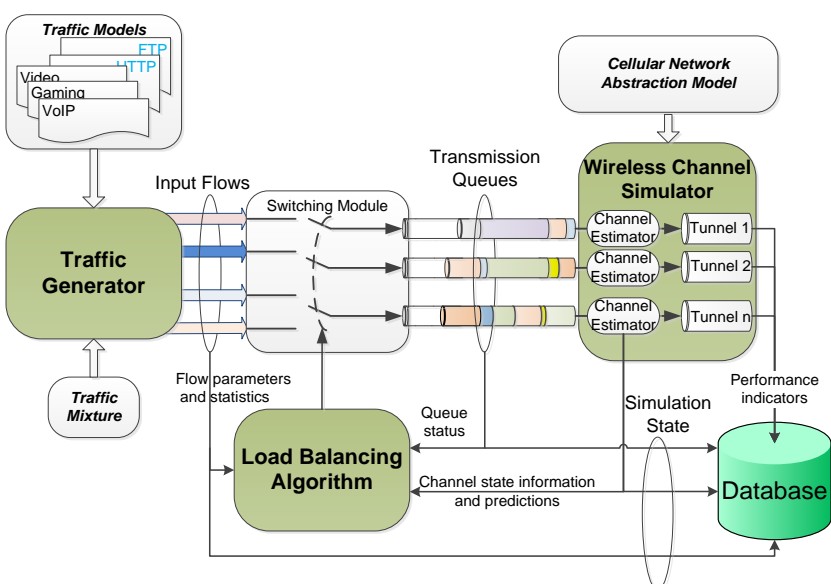

**Figure 6.** The block diagram of the simulation platform used for the evaluation of the LB algorithms.

3.1.1. The Traffic Generator Module

This module generates the data packets which constitute the input traffic for the simulation chain. The packets are generated based on statistical models used for system level evaluation of the cellular wireless systems [43–45]. The traffic types considered are: Best Effort Traffic (FTP), Interactive Traffic (Web-browsing using HTTP), VoIP, Video Streaming, and Interactive Real-Time Services (Gaming). All the traffic flows are generated according to the specifications of [43–45], the most relevant parameters being considered for each traffic flow. See Appendix A for the mathematical modeling of the mentioned traffic types.

3.1.2. The Wireless Channel Simulator Module

The cellular network scenario considered in the design of the wireless channel simulator is presented in Figure 1. The scenario is a realistic one that supposes the overlapping of the coverage area of several base stations using the same or different wireless access technologies. A device, i.e., a mobile router or a mobile AP, equipped with $N$ cellular network interfaces is moving in an area covered by $N$ independent wireless networks. The device follows a random walk motion pattern, and when the device arrives at the border of the cell, the position of the new serving base station is generated randomly so that the radius of the cells follows a Gaussian distribution with predefined parameters. In this case, we obtain a realistic network topology where the position of the base stations differs from one operator to another as well as the size of the cells the user passes through while moving on his route. It is also assumed that all networks serving the device have approximately the same load, which is less than their capacity, and each network uses a proportional fair radio resource management algorithm. In this case, the average transfer rate obtained by the device is highly related to the radio channel characteristics experienced by that device. In this simulator, it is considered that the instantaneous transfer rate is proportional with the effective instantaneous Signal to Noise Ratio (SNR) observed at the input of the considered receiver. The considered effective SNR–transfer rate mapping model is depicted in Figure 7. A modified ITU Vehicular B [46] channel model was considered, with mean mutual information per coded bit (MMIB)-based link to the system mapping technique.

As an example, a particular situation regarding the capacity variation of the virtual tunnels established between the AP and the four base stations depicted in Figure 1 is presented in Figure 8. In Table 1 are presented the max./min. values and the mean, median, and standard deviation values of the transfer rates of the 4 considered tunnels. Can be noticed that some of the tunnels provide relatively constant transfer rates (tunnels 2 and 4), while on other tunnels the transfer rates variers significantly in time (tunnels 1 and 3).

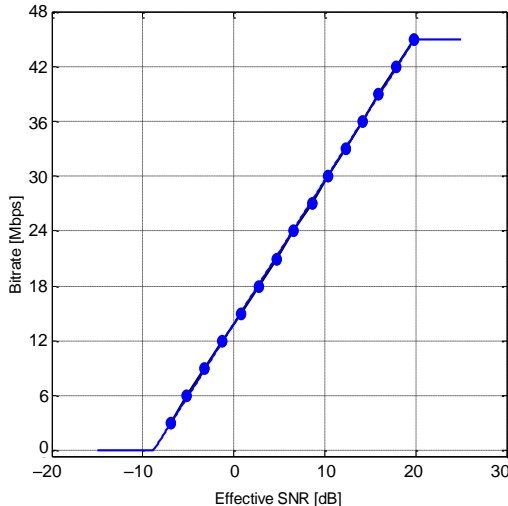

**Figure 7.** Effective SNR –bitrate mapping.

**Table 1.** Statistical parameters of the considered 4 virtual tunnels.

| Tunnel | Max/Min | Mean | Median | Std. |
|---|---|---|---|---|
| 1–red | 24/14.5 Mbps | 20.15 Mbps | 21.25 Mbps | 2.85 Mbps |
| 2–blue | 18.5/14.25 Mbps | 16.5 Mbps | 16.5 Mbps | 1.15 Mbps |
| 3–cyan | 14.75/1.5 Mbps | 6.5 Mbps | 5.75 Mbps | 4 Mbps |
| 4–green | 8.5/5.25 Mbps | 6.75 Mbps | 6.75 Mbps | 0.925 Mbps |

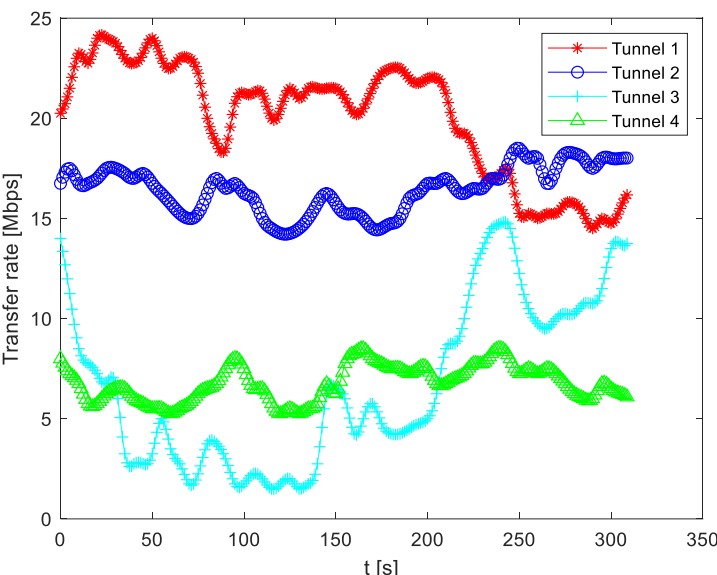

**Figure 8.** Variation in time of the capacity of the tunnels established over the macro cell wireless links.

### 3.2. The Architecture of the Simulation Platform Used for the Evaluation of the User Clustering Algorithms

To test the proposed user clustering algorithms another simulation platform, depicted in Figure 9, was developed. The integration of the users' clustering algorithms in the complex simulation platform used for the evaluation of the presented LB algorithms is relatively difficult to perform. The developed platform includes the data flows generation module (User Cluster Traffic Generator), based on the considered traffic models (see Tables A1–A5) and traffic mixture specified in Table 2. To simplify the simulation a single output tunnel is considered, and this could be an aggregated tunnel. This aggregated tunnel is simulated by the Wireless Channel Simulator module. An AP Controller module is integrated into the simulation platform. This module receives the aggregated tunnel queues' status information and the data flows' characteristics and controls the transmission power of the APs according to the used clustering algorithm. Based on the selected transmission power the user clusters connected to different APs are generated, more users connecting to the APs with larger power. This approach has the target to simplify the users' access to the APs, only the power of the APs being manipulated. The APs with more available resources increase their transmission power on the AP-user links, while the APs with fewer available resources maintain or reduce their transmission power. AP selection or handover based on the received power level is one of the most common and simplest ways of selecting an attachment point to a wireless network and this solution is used on large scale also in cellular networks.

**Table 2.** Simulated traffic flows parameters.

| Traffic Type | Traffic Mixture | LB Parameter |
|---|---|---|
| VoIP | 20% | |
| Video streaming | 20% | $c_i = 5\text{ms}$ |
| Online gaming | 10% | |
| Web browsing-HTTP | 30% | $R_i^{av} = 10\text{Mbps}$ |
| File transfer-FTP | 20% | |

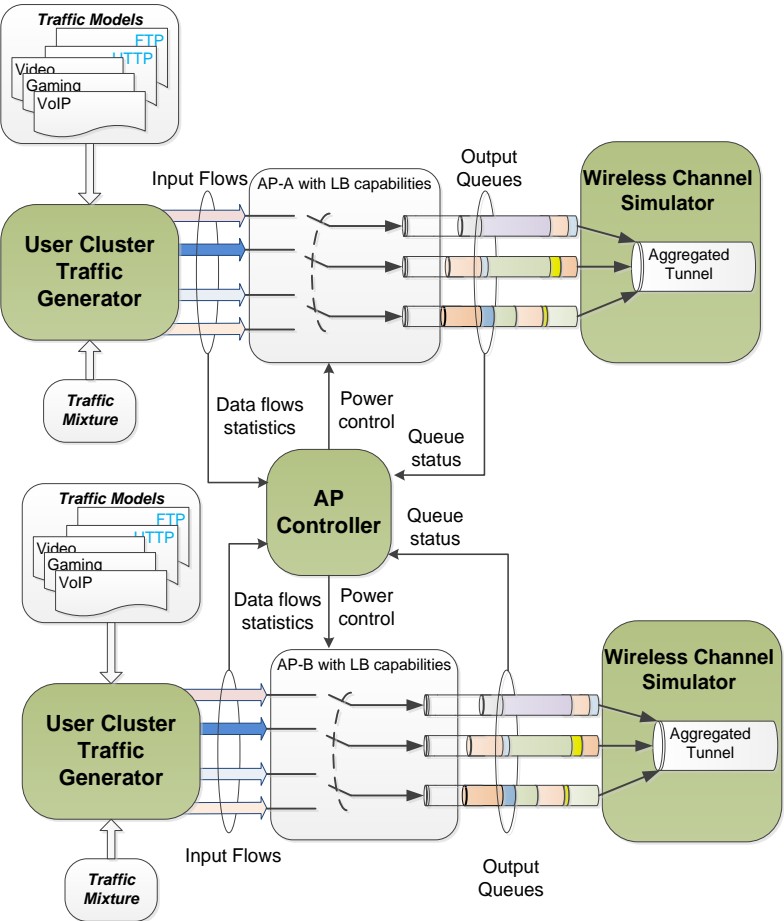

**Figure 9.** The block diagram of the simulation platform used for the evaluation of the user clustering algorithms.

The users are randomly distributed between the deployed APs and the distances between the users and each AP are computed. Based on the distances the attenuation on the user-AP links and the power received by the users are computed according to Equation (29).

$$a_{user-AP}(d)[dB] = a_s(d_0)[dB] + 10 \cdot n \cdot \lg \frac{d}{d_0}$$

$$a_s(d_0) = \left( \frac{4\pi \cdot d_0}{\lambda} \right)^2 \tag{29}$$

$$P_{user} = P_{AP} - a_{user-AP}$$

where $d_0$ is the reference distance and is considered 1 m (it depends on the size of the cell), $n$ gives the slope of the attenuation characteristic and is considered 2.5, but it depends on the simulated wireless propagation scenario and $\lambda$ is the wavelength–the carrier frequency was considered 2.4 GHz.

The estimation of the transmission resources available on the virtual tunnels can be made based on the sojourn time $S$, i.e., the delays suffered by the source data packets in the transmission system. To take into account the variation of the sojourn time on a larger time scale, a moving average is computed on each tunnel considering a large number of packets $N$, i.e., several thousands of packets:

$$S_{av\_i} = \frac{S_i + N \cdot S_{av\_i-1}}{N+1} \tag{30}$$

where $S_{av\_i}$ is the average delay at moment *i*.

Other indicators related to the available resources in the system are the number of packets stored in each buffer and the number of bytes (or bits) stored in the same buffers. These parameters combined with the sojourn time could give a more detailed picture of the available resources and allow a better clustering of the users.

The clustering process of the users should be performed with a relatively low frequency to avoid frequent disconnection and reconnection of the users (the so-called 'ping-pong' effect) which could generate interruption in the transmission process and increase the signaling overhead. However, at the same time the clustering process should be performed frequently enough to take into account the variability of the capacities of the tunnels and the users' data flows rates. In the simulations performed the clustering process was performed at a time interval of tens of seconds, but less than 1 min. The signaling overhead on the user-AP link was not considered.

### 3.3. Simulation Results and Discussions

3.3.1. Evaluation of the Proposed LB Algorithm over Parallel Virtual Tunnels

Four tunnels are set up in four different cellular networks in the evaluation scenario considered (see Figure 1), and 50 data flows are generated by the users connected to the small cell AP. We consider that the cellular networks belong to different operators, and they are independent and assign uncorrelated transfer rates to the tunnels. The considered channel and shadowing model is the urban environment model defined in [46]. The considered traffic mixture [44] of the data flows is defined in Table 2. The performance indicator which is monitored is the evolution in time of the delays suffered by the packets of the transmitted flows when different LB algorithms are used, i.e., the proposed LB algorithms and the considered reference LB algorithms. The used traffic mixture is a relevant and balanced one with 50% percent real-time traffic and 50% best effort traffic. A high percentage of real-time traffic will not be relevant for the assessment of the developed LB algorithm due to the relatively low bit rates of these traffic types, which will generate an underuse of the tunnels' transfer capacities and the delay requirements will be fulfilled without any problems. A high percentage of best effort traffic, with a relatively large expected average rate $R_i^{av}$, will generate an overload of the tunnels and an increase in the delays inserted by the transmission system, no matter which LB algorithm is used.

Figure 10 presents the Probability Density Functions (pdf) (Figure 10a) and the Cumulative Density Functions (cdf) (Figure 10b) of the delays suffered by the transmitted packets in the simulated transmission system when the proposed, GT-based LB algorithms, respectively when the reference LB algorithms are used.

As expected, the delays experienced by the transmitted packets are the largest in the case of the Round Robin LB algorithm. This algorithm does not consider the statistics of the data flows and the quality of the transmission links over which the tunnels are instantiated. Due to this, the requested transfer rate on a given tunnel could become larger than the instantaneous capacity with a high probability, and this will lead to large delays of the packets waiting in the queues.

Using the Selfish Routing LB algorithm each packet should choose a tunnel where the expected delay is minimal but the frequent tunnel changes generate a high amount of signaling traffic. In this case, the number of the generated signaling packets was comparable with the number of payload packets, and this high overhead overloads the tunnels leading to significant delays. In the scenario when the signaling overhead is neglected (or at least significantly reduced) the experienced packet delays are much lower than in the previous case when the signaling overhead is considered. Even in this case, the social cost of the Nash equilibrium is higher than that of the optimal solution due to the lack of coordination (the price of anarchy). This loss in performance relative to the optimal solution is because in each time instance more packets could choose a given tunnel where apparently the delay conditions are the best, overloading the tunnel, thus generating temporary

congestion on it, while other tunnels have free capacity. Reducing the signaling overhead can be achieved by routing several consecutive packets on the same tunnel, i.e., dealing with flows instead of individual packets.

The Auction-based LB algorithms ensure the best performance because they dynamically adapt the traffic on the tunnels according to the changes in the transmission links parameters. The Combinatorial Auction based LB algorithm over-performs the other considered algorithms because it permanently monitors the QoS performance and the system's changes, and it maximizes both the user's and the system's revenue (measured by the level of satisfaction or the cost function). The Auction-based LB algorithm also has good performance, the difference between the two GT-based algorithms not being significant, at least at low packet delays.

The reference Multiple Knapsack algorithm has lower performance than the auction-based algorithms and ensures comparable performance with the ideal Selfish Routing LB algorithm with no or low overhead.

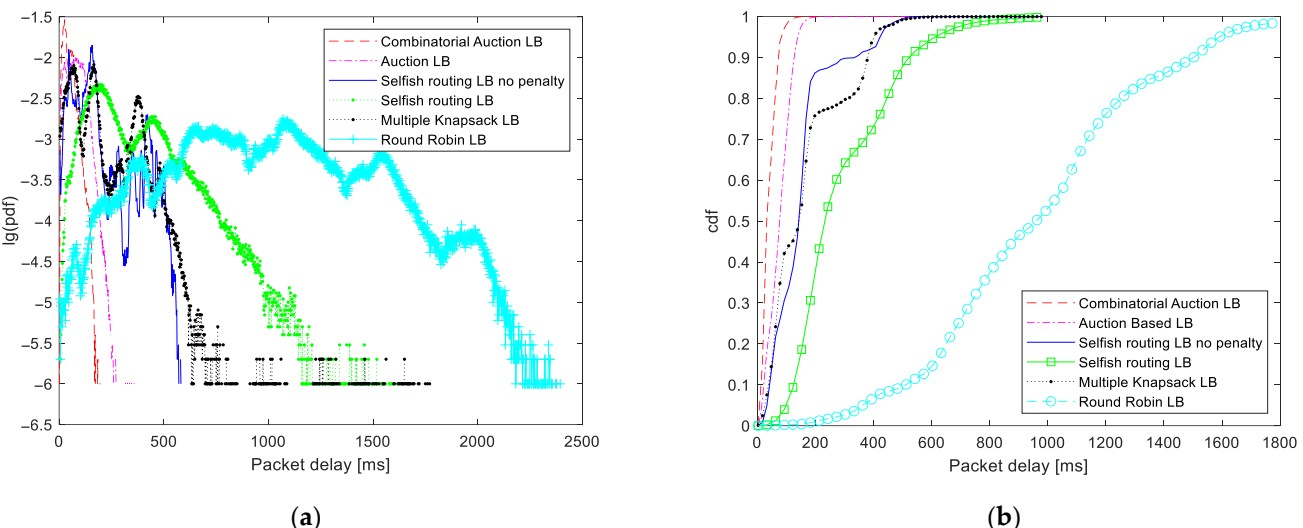

(**a**)                                                                                     (**b**)

**Figure 10.** Statistical characterization of the packet delays inserted by the considered LB algorithms: (**a**) The pdfs of the packet delays; (**b**) The cdfs of the packet delays.

Figure 11 presents separately the pdfs and the cdfs of the delays suffered by the transmitted packets in the simulated transmission system when the Auction-based LB algorithm, the Combinatorial Auction-based LB algorithm and the Selfish Routing LB algorithm with no penalty are used, respectively when the reference Multiple Knapsack LB algorithm is used (these are the best algorithms identified in Figure 10). The presented results show clearly that the Combinatorial Auction algorithm has the best performance, i.e., it inserts the smallest delays with the highest probabilities, and the Auction-based LB algorithm also has good performance. Both algorithms have significantly better performance, i.e., smaller delays with higher probabilities and larger delays with lower probabilities, compared to the Multiple Knapsack algorithm and the Selfish Routing LB algorithm with no signaling penalty. Even if all algorithms have polynomial complexity, the highest processing resources are requested by the Combinatorial Auction-based algorithm followed by the Multiple Knapsack and the Auction-based LB algorithms. The Combinatorial Auction-based algorithm has the highest computational complexity due to the continuous monitoring of the potential functions and the instantaneous call throughput, while the other two algorithms (mentioned above) must evaluate only the instantaneous call throughput.

Another important issue that must be considered is how well can handle the proposed and tested LB algorithms the real-time flows. The distributions of the packet delays presented in Figures 10 and 11 do not make any difference between the different traffic

types, being characterized the entire traffic mix. In the case of the real-time type data flows (see Appendix A) the interpacket delays are typically situated between 0 and 40–60 ms. In the case of VoIP flows the interpacket delay is 20 ms, the maximum allowed value of interpacket delays for video traffic is 12.5 ms and for gaming the interpacket delays are requested to be smaller than 40 ms, a 60 ms delay being considered an outage. The values of the packet delay cdfs of the algorithms considered in Figure 11 are presented in Table 3, for a better comparison of these values.

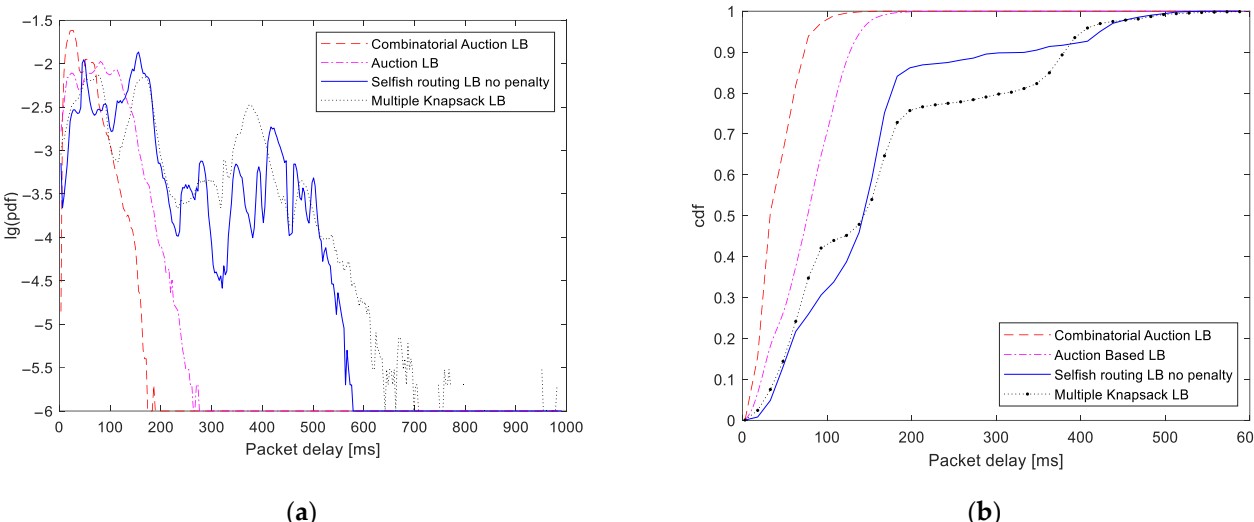

(**a**)            (**b**)

**Figure 11.** Statistical characterization of the packet delays inserted by the bests of the considered LB algorithms: (**a**) The pdfs of the packet delays; (**b**) The cdfs of the packet delays.

**Table 3.** Values of the packet delay cdfs for the LB algorithms considered in Figure 11.

| LB Algorithm | cdf Function/Packet Delay (ms) | | | |
|---|---|---|---|---|
| Combinatorial Auction | 0.23/20 ms | 0.58/40 ms | 0.8/60 ms | 0.95/80 ms |
| Auction | 0.08/20 ms | 0.22/40 ms | 0.36/60 ms | 0.54/80 ms |
| Selfish routing no signaling penalty | 0.01/20 ms | 0.09/40 ms | 0.21/60 ms | 0.28/80 ms |
| Multiple Knapsack | 0.025/20 ms | 0.1/40 ms | 0.21/60 ms | 0.38/80 ms |

The results presented in Table 3 show clearly that only the Combinatorial Auction-based LB algorithm can ensure with high probability (in the considered scenario) the delays requested by the considered real-time applications. The Combinatorial Auction-based LB algorithm computes the potential functions for each traffic flow and checks continuously if the QoS parameters can be ensured on each tunnel and if not, the data flows are redistributed. The Selfish Routing and the Auction-based LB algorithms do not consider separately the different types of flows and do not check the fulfillment of the QoS requirements after the routing decision is taken. The Multiple Knapsack LB algorithm assigns different priorities for different types of flows and routes first the flows with higher priority, but it takes the routing decisions only based on average flow rates and average tunnel capacities. The results presented in Figure 8 and Table 1 show that the wireless tunnels' capacities could change significantly in time having as consequence the possible (significant) increase of the sojourn time.

The results presented above are valid in the situation when the total transfer rate of the users' flows is less than the total capacity of the communication tunnels, but the difference between the requested rate and the offered capacity is relatively small when it is important to fill each tunnel as close to capacity as possible. The selected flow mixture represents such a situation. If this difference is large, it does not matter which LB algorithm is used, the tunnels being capable of accommodating the source flows.

The analysis presented above shows that if the tunnels are congested and real-time applications are used by many users the Combinatorial Auction algorithm is the best solution, this algorithm being capable to fulfill with high probability the latency requirements of real-time applications. If the level of congestion on the tunnels is reduced the simple Auction-based algorithm is the best choice having better performance than the Multiple Knapsack algorithm but lower implementation complexity. Due to the lack of control, the Selfish Routing algorithm is not a practical option, the Multiple Knapsack algorithm being the best solution if the auction-based GT algorithms cannot be used.

Figure 12 presents the price of anarchy, $max_{NE}$ (20), for the four tunnels considered in the performed experiment. Figure 12a presents the evolution of the price of the anarchy with the absolute value of the total rate $R_{max} < R_{av\text{-}tunnel}$, for each tunnel, while in Figure 12b the variation of the price of anarchy with the relative value of the total rate $R_{max}/R_{av\text{-}tunnel}$ is presented.

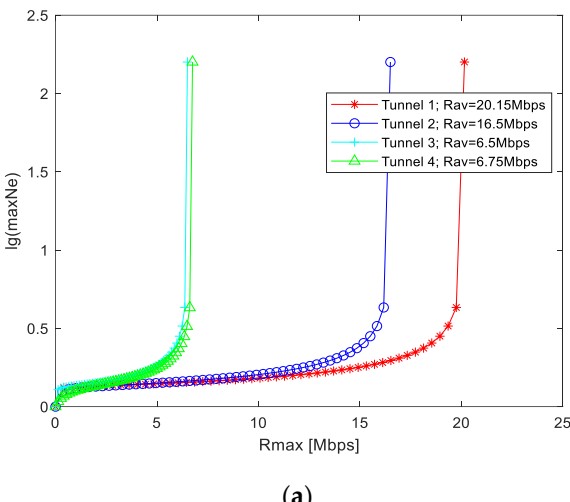
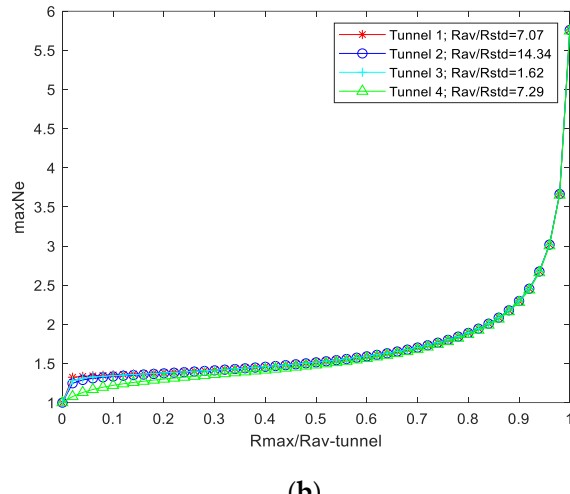

(**a**)                    (**b**)

**Figure 12.** Price of anarchy, *max$_{NE}$*, for the simulated virtual tunnels. (**a**) *max$_{NE}$* versus the absolute total rate *R$_{max}$*; (**b**) *max$_{NE}$* versus the relative total rate *R$_{max}$/R$_{av-tunnel}$*.

The price of anarchy increases abruptly when the total rate *R$_{max}$* approaches the average transfer rate of the tunnel, but for values of *R$_{max}$* smaller than *R$_{av-tunnel}$* the variation of the price of anarchy is relatively slow with the increase of *R$_{max}$* (see Figure 12a). Furthermore, we can see that as the total rate *R$_{max}$* approaches the *R$_{av-tunnel}$* value of the tunnel transfer rate significant gains in system transfer rate can be obtained by using more efficient LB algorithms than the relatively simple selfish routing.

If we compare the variation of the price of anarchy for the considered tunnels with the values of the *R$_{max}$/R$_{av-tunnel}$* ratio (the relative total rate) results that the variation of the price of anarchy has (approximately) the same behavior no matter the average rate/std rate ratio, *R$_{av-tunnel}$/R$_{std-tunnel}$*, of the tunnel (see Figure 12b).

### 3.3.2. Practical Implementation Issues

The proposed and discussed GT-based and reference LB algorithms act on packet/flow level which makes possible a network layer implementation. This is a significantly simpler solution than a MAC/Physical layer implementation, even if the latter one has the potential of a more efficient implementation. To implement the LB algorithms at the network layer access is required to various packet queues embedded in the Operating System (OS) kernel. Packet queues are basic components of any network stack. Queues allow communication between asynchronous modules, and can increase the performance of inter-module communication, but have the side effect of impacting latency. In Figure 13 is depicted the enqueuing of the IP packets on the transmit path of the Linux OS network stack, being highlighted also some of the used latency-reducing features [47].

Between the IP stack and the Network Interface Controller (NIC) is located the NIC's driver queue (see Figure 13), typically implemented as a FIFO buffer. The driver queue does not necessarily store the packet data. An alternative is to store some descriptors that point to other packet queues called socket kernel buffers (SKBs), where the packets of the service flows are stored. The NIC's driver queue acquires packets from the IP stack. These packets may be generated locally by various services running on the platform or may be received on one NIC and routed to another one when the device is functioning as an IP router. The use of the driver queue ensures that if the system has data packets to transmit these packets will be sent on the channel at the moment when the NIC becomes ready for transmission. By employing this design, the NIC does not have to ask the IP stack for data packets when transmission opportunities occur, resulting in a higher transfer rate.

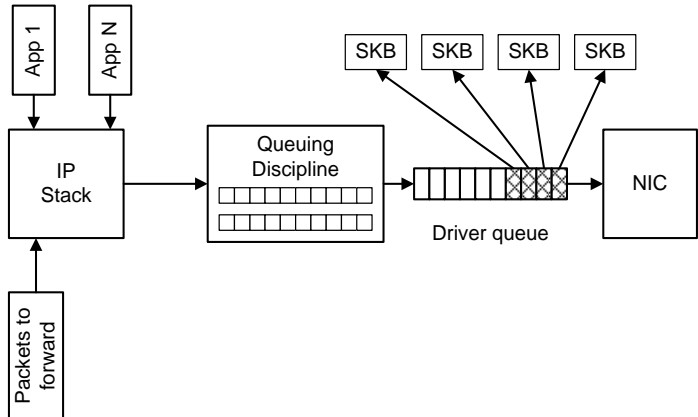

**Figure 13.** A simplified high-level overview of the queues on the transmit path of the Linux OS network stack.

The discussed LB algorithms could be integrated into the Queuing Discipline module, which sets the order in which different services can access the driver queues of the NIC. The Queuing Discipline module has access to the IP stack queues and can acquire information about the data flows. This module also has access to the NIC's driver queues and can acquire information about the delay inserted by each tunnel (each tunnel is attached to a separate NIC) and the number of packets waiting in the NIC's queues. The LB algorithms discussed can also be run in the user space but will be necessary to use functions that allow access from the user space to the mentioned queues. It should also be mentioned that the GT-based LB algorithms proposed can obtain all the parameters necessary for performing the LB operations by analyzing the evolution in time of the queues mentioned above. The identification of the service flows and the measurement of packet rates or the interpacket delays can be carried out by analyzing the content and evolution in time of the IP stack queue. The measurement of the delays suffered by various packets and the congestion level on the communication tunnels implementing the backhaul connections can be achieved by analyzing the evolution in time of the NICs' transmit queues. No other measurement of the macro cell wireless links parameters is necessary, which simplifies the system, not being necessary to use specific measurement techniques for different macro cell links technologies. Moreover, the change of the macro cell wireless links characteristics and/or parameters will not require changes in the LB algorithms.

### 3.3.3. Evaluation of the Proposed User Clustering Algorithms

The evaluation of the user clustering algorithms was performed in the same conditions as the evaluation of the LB algorithms, meaning that the number of users and the traffic mix considered is the same. The scenarios considered are those presented in Figure 4, i.e., the scenario with the deployment of 2 APs and the scenario with the deployment of 3 APs. The APs are located at fixed distances $D$, $D$ = 100 m in the simulations performed, and the users are spread uniformly between the 2 or the 3 APs. The traffic types are randomly distributed among the users, but the imposed traffic mix is maintained (see Table 2). To simplify the simulations, it is considered that to each AP it is attached a single virtual tunnel, which could be the aggregation of several separate virtual tunnels. The transfer rate of these aggregated tunnels originating in the APs was simulated as in the case of the evaluation of the LB algorithms, the same traffic models being used. The parameters of the channel models selected to obtain approximately the same average transfer rate on each aggregated tunnel (approximately 20 Mbps). The step used to adjust the power of the APs was set to 1 dB in some experiments and 2 dB in other experiments, but no significant difference can be identified between the results obtained in these two cases. In all scenarios, the statistics of the delays suffered by the transmitted packets are

evaluated. Figure 14 presents the pdf of the packet delays in the system (the sojourn time) in the scenario with two APs and the following cases:

- the two APs work separately, and no user clustering algorithm is used. The clustering of the users is performed only at the beginning of the simulation by attaching each user to the AP from which it receives the largest power–see Figure 14a.
- a non-cooperative type of user clustering is used, each AP adjusting its transmission power only based on its local QSI and CSI–see Figure 14b.
- a cooperative user clustering algorithm is used, each AP having access to the QSI and CSI of other APs or being controlled by an AP Controller who has access to all APs' QSI and CSI. If the controller is used this module implements the AP cooperation process and sets the transmission power of each APs–see Figure 14c.

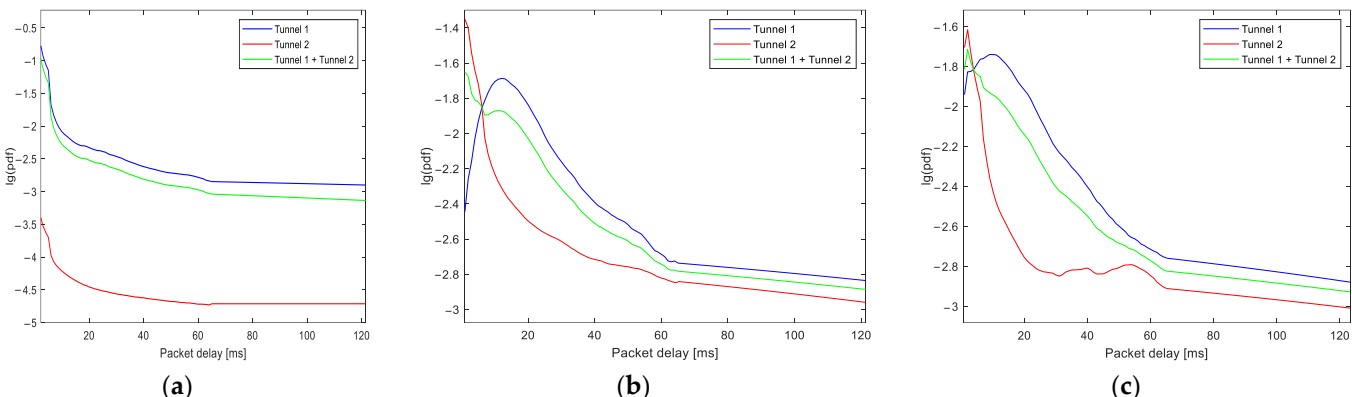

**Figure 14.** Statistical characterization of the packet delays on the aggregated virtual tunnels when 2 APs and different user clustering algorithms are used: (**a**) The pdfs of the packet delays when no user clustering algorithm is used; (**b**) The pdfs of the packet delays when the non-cooperative user clustering algorithm is used; (**c**) The pdfs of the packet delays when the cooperative user clustering algorithm is used.

Figure 14 presents separately the pdf of the delay suffered by the transmitted packets on the two tunnels and the pdf of the delays suffered by all the packets transmitted by the system, meaning the pdf on the combined tunnel used by the system.

The results presented in Figure 14 show the effects of the LB performed by the clustering algorithms among the APs. If no clustering is used most of the users might be attached to one of the APs, one of the outgoing aggregated tunnels being heavily loaded (red curve in Figure 14a) while the other tunnel (the blue curve in Figure 14a) is underused. If any of the clustering algorithms are used the difference in the traffic loads on the two aggregated tunnels becomes significantly smaller. To better assess the statistics of the delays suffered by the data packets on the two aggregated tunnels in the considered situations the cdf of the packet's delays is presented in Figure 15.

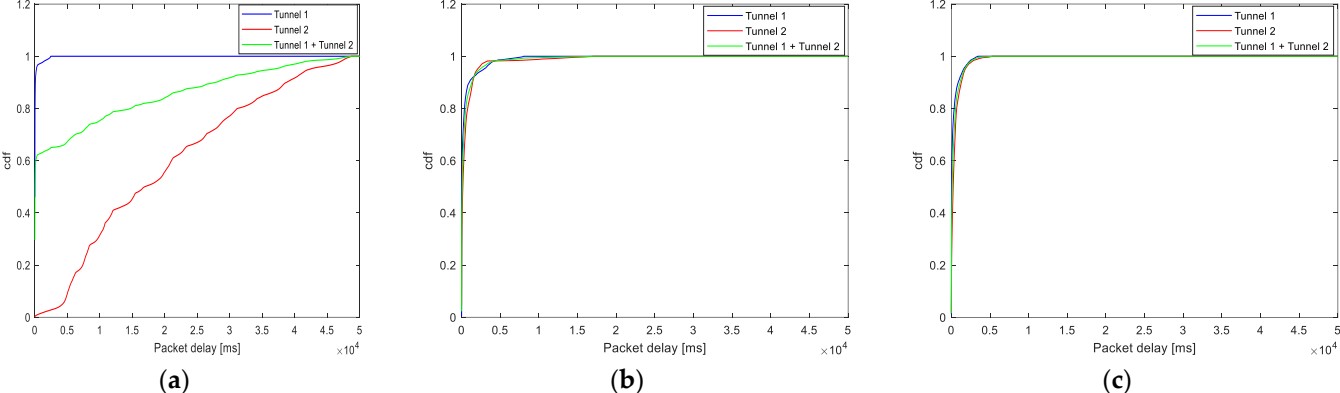

**Figure 15.** The Cumulative Density Functions of the packet delays on the aggregated virtual tunnels when 2 APs and different user clustering algorithms are used: (**a**) The cdfs of the packet delays when no user clustering algorithm is used; (**b**) The cdfs of the packet delays when the non-cooperative user clustering algorithm is used; (**c**) The cdfs of the packet delays when the cooperative user clustering algorithm is used.

The results presented in Figure 15 show clearly the huge difference in performance, in what concerns the packet delays, of the systems with no users clustering (Figure 15a) and with user clustering (Figure 15b,c). Comparing Figure 15b,c we can see the difference between the non-cooperative and the cooperative clustering algorithms. In Figure 15c the curves converge around 500 ms while in Figure 15b the curves converge around 1500 ms. Still, should be noticed that the differences in the performance depicted in Figure 15b,c, i.e., the distribution of the packet's delays, are small.

Figure 16 presents the pdf of the packet delays in the system (the sojourn time) in the scenario with three APs and the following cases:

- the three APs work separately, and no user clustering algorithm is used. The clustering of the users is performed only at the beginning of the simulation by attaching each user to the AP from which it receives the largest power–see Figure 16a.
- a cooperative user clustering algorithm is used, and the APs power adjustment step is set to 1 dB–see Figure 16b.
- a cooperative user clustering algorithm is used, and the APs power adjustment step is set to 2 dB–see Figure 16c.

A cooperative user clustering algorithm with separate a power adjustment step for the central AP (1 dB) and for the side APs (0.5 dB) was also tested in the same conditions, but the results are very similar to those presented in Figure 16b, i.e., 1 dB power adjustment step. In Figure 16 are presented the pdfs of the packet delays on each tunnel and the combined tunnel used by the simulated system.

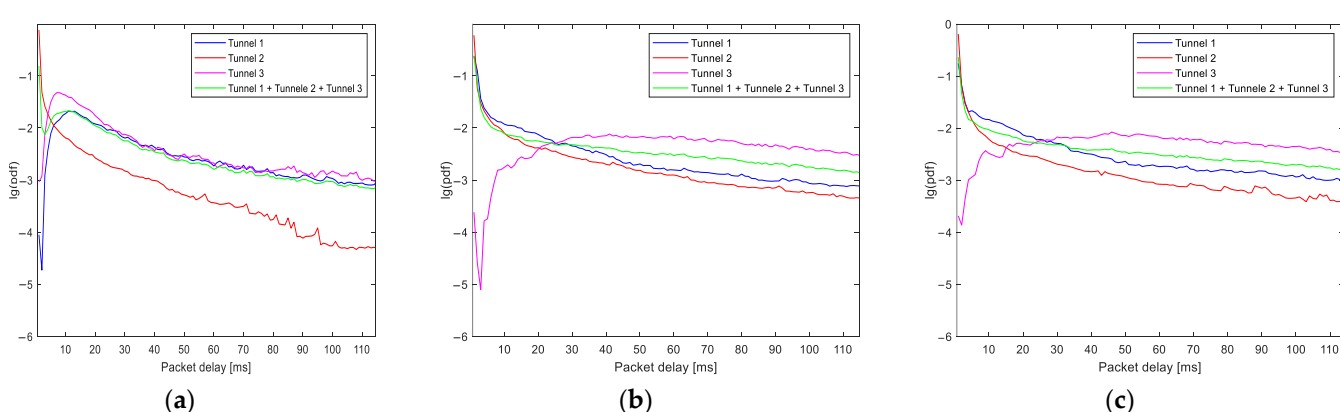

**Figure 16.** Statistical characterization of the packet delays on the aggregated virtual tunnels when 3 APs and different user clustering algorithms are used: (**a**) The pdfs of the packet delays when no user clustering algorithm is used; (**b**) The pdfs of the packet delays when the cooperative user clustering algorithm with 1 dB power adjustment step is used; (**c**) The pdfs of the packet delays when the cooperative user clustering algorithm with 2 dB power adjustment step is used.

In Figure 16a, when no clustering algorithm is used, there is a significant difference between the load of tunnel 2 (red curve) and tunnels 1 and 3 (blue and magenta curves), these two last tunnels being more loaded, especially tunnel 1 (blue curve). If the clustering algorithms are used, the load of the APs and the outgoing tunnels attached to the APs will change, and the load of tunnel 1 will decrease significantly, its load being distributed between tunnel 2 and 3 which will become more loaded, especially tunnel 3. Should be noticed that the performance of the clustering algorithm depends significantly on the position of the users and the characteristics of the service flows generated by the users and an optimal (more exactly close to optimal) distribution of the traffic between APs is not possible without using more complex clustering algorithms with more complex interaction between the APs and the users. The difference between the load of the APs in the 3 situations mentioned above is more visible if we analyze the cdfs of the packet delays depicted in Figure 17. In Figure 17a all curves converge around 3500 ms, while in Figure 17b,c all curves converge around 1500 ms. The performance of LB among the APs for power adjustment step 1 dB and 2 dB are very close.

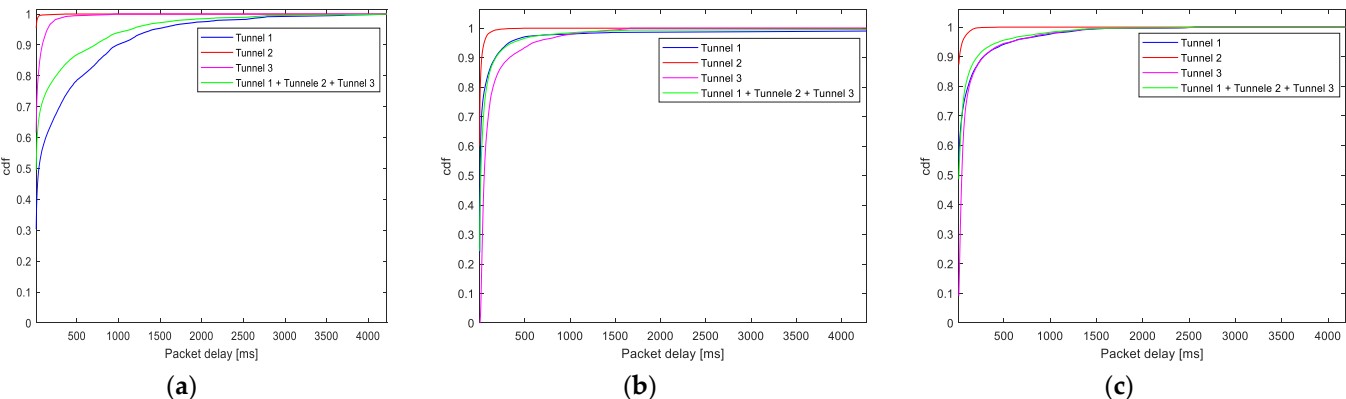

(**a**)             (**b**)             (**c**)

**Figure 17.** The Cumulative Density Functions of the packet delays on the aggregated virtual tunnels when 3APs and different user clustering algorithms are used: (**a**) The cdfs of the packet delays when no user clustering algorithm is used; (**b**) The cdfs of the packet delays when the cooperative user clustering algorithm with 1 dB power adjustment step is used; (**c**) The cdfs of the packet delays when the cooperative user clustering algorithm with 2 dB power adjustment step is used.

It should be noticed that the clustering algorithms, like the LB algorithms acting on the backhaul communication tunnels, require access only to the queue of the IP stack and the driver of the NIC that provides the AP-user connectivity. Identification of the data flows and the measurement of the parameters of these flows can be achieved by analyzing the content of the IP stack queue while the load of the AP-users connection can be assessed by analyzing the evolution in time of the length of the NIC's driver queue.

## 4. Conclusions

The paper considers the issue of LB over several wireless links set up in a heterogeneous cellular system, wireless links which are used as backhaul connections for the AP of a small cell. The paper proposes game theory-based algorithms for distributing the data flows generated by the users connected to the AP of the small cell over the wireless links of several macro cells, possibly owned by different operators, which cover the small cell. More precisely the Selfish Routing LB algorithm, the Auction-based LB, and the

Combinatorial Auction-based LB algorithms are proposed. As references, the classical Round Robin algorithm and an algorithm based on the Multiple Knapsack problem were considered. Computer simulations performed in a complex scenario involving four macro cells and a mobile AP, i.e., installed in a vehicle, and in the condition of a congested network show that the Auction-based LB and the Combinatorial Auction-based LB algorithms have the best performance, while the performance of the Selfish Routing LB algorithm is weaker than that of the reference Multiple Knapsack LB algorithm, or is similar with the performance of this reference algorithm if the signaling penalty is not considered.

The simulations show that in the condition of congested networks the Combinatorial Auction-based LB algorithm and the Auction-based LB algorithm are capable to manage efficiently the available resources, i.e., to distribute efficiently the flows on the available tunnels, and the packets' delays can be kept below some limits. The Multiple Knapsack LB algorithm can manage well the LB operations for some reduced traffic values, and reduced packet delays are ensured with probabilities close to that obtained with the Auction-based LB algorithm. In the situation of larger (instantaneous) traffic values the performance of the Multiple Knapsack LB algorithm decreases, and larger delays appear with large probabilities, meaning that the LB operations are performed less efficiently compared to the GT-based LB algorithms. The Round Robin LB algorithm has, as expected, poor performance, large (even very large) packet delays appearing with high probabilities.

The proposed GT-based LB algorithms are not constrained only to the data communication scenario considered by this paper. The algorithms are general enough to be adapted to other communication scenarios, involving both wireless and wired connection, when LB operations are necessary to distribute data flows over parallel communication channels/tunnels.

The paper also proposes two game theory-based algorithms for user clustering when several small cell APs are deployed. These algorithms select the users who connect to each AP, the goal being to avoid the overloading of some APs while other neighbor APs have available resources. The proposed algorithms use a simple power adjustment-based user clustering to avoid more complex interaction and signaling between the users and the APs. The performed simulations show that even in the case of simple clustering mechanisms the load could be distributed relatively well between the deployed APs and therefore a noticeable decrease in the packets' delays on the virtual tunnels attached to the APs is visible.

**Author Contributions:** Conceptualization, Z.A.P. and M.V.; methodology, Z.A.P.; software, M.V.; validation, Z.A.P. and M.V.; formal analysis, Z.A.P.; investigation, M.V.; resources, Z.A.P.; writing—original draft preparation, Z.A.P.; writing—review and editing, Z.A.P. and M.V.; visualization, M.V.; supervision, Z.A.P. All authors have read and agreed to the published version of the manuscript.

**Funding:** This research received no external funding.

**Institutional Review Board Statement:** Not applicable.

**Informed Consent Statement:** Not applicable.

**Data Availability Statement:** Data sharing not applicable.

**Conflicts of Interest:** The authors declare no conflict of interest.

## Appendix A Data Traffic Modeling

*Appendix A.1. Best Effort Traffic: FTP*

An FTP session is composed of a succession of file transfers separated by some time intervals, the so-called reading time. The main FTP session parameters are (see Table A1):

- The size S of a file to be transferred;

- The reading time D, i.e., the time interval between the end of the download of the previous file and the beginning of the next file transfer.

Based on experimental evaluation of the FTP sessions was observed that for each file transfer a separate TCP connection is used and that around 76% of the files were transferred using a Maximum Transmission Unit (MTU) of 1500 bytes and around 24% of the files were transferred using an MTU of 576 bytes.

**Table A1.** FTP traffic parameters.

| Parameter | Statistical Characterization |
|---|---|
| File Size $S$ | Truncated Lognormal Distribution<br>Mean = 2 Mbytes, Standard Deviation = 0.722 Mbytes, Maximum = 5 Mbytes (before truncation)<br>pdf: $f(x) = \dfrac{1}{\sqrt{2\pi}\sigma x} e^{\frac{-(\ln x - \mu)^2}{2\sigma^2}}$ , $x > 0$, $\sigma = 0.35$, $\mu = 14.45$ |
| Reading Time $D$ | Exponential Distribution<br>Mean = 180 s<br>pdf: $f(x) = \lambda e^{-\lambda x}$　$x \geq 0$, $\lambda = 0.006$ |

*Appendix A.2. Interactive Traffic: Web Browsing Using HTTP*

A web page is formed of a main object and a set of embedded objects (e.g., pictures, advertisements, etc.). After the main page is received, the web browser parses the embedded objects. The main parameters that characterize the HTTP web browsing are (see Table A2):

- The main size of an object $S_M$;
- The size of an embedded object in a page $S_E$;
- The number of embedded objects $N_D$;
- Reading time $D$;
- Parsing Time for the embedded page $T_P$;

**Table A2.** Web-browsing traffic parameters.

| Parameter | Statistical Characterization |
|---|---|
| Main Object Size $S_M$ | Truncated Lognormal distribution. Mean = 25,032 bytes, Standard Deviation = 10,710 bytes, Minimum = 100 bytes, Maximum = 2 Mbytes (before truncation)<br>pdf: $f(x) = \dfrac{1}{\sqrt{2\pi}\sigma x} e^{\frac{-(\ln x - \mu)^2}{2\sigma^2}}$ , $x > 0$, $\sigma = 1.37$, $\mu = 8.37$ |
| Embedded Object Size $S_E$ | Truncated Lognormal distribution<br>Mean = 126,168 bytes, Standard Deviation = 7758 bytes, Minimum = 50 bytes, Maximum = 2 Mbytes (before truncation).<br>pdf: $f(x) = \dfrac{1}{\sqrt{2\pi}\sigma x} e^{\frac{-(\ln x - \mu)^2}{2\sigma^2}}$ , $x \geq 0$, $\sigma = 2.36$, $\mu = 6.17$ |
| Number of Embedded Objects per Page = $N_D$ | Truncated Pareto distribution<br>Mean = 5.64, Maximum = 53 (Before Truncation)<br>pdf: $f(x) = \dfrac{\alpha_k^{\alpha}}{\alpha+1}, k \leq x < m$ , $f(x) = \left(\dfrac{k}{m}\right)^{\alpha}, x = m$ , $\alpha = 1.1$, $k = 2$, $m = 55$ |
| Reading Time $D$ | Exponential Distribution, Mean = 30 s<br>pdf: $f(x) = \lambda e^{-\lambda x}, x \geq 0$, $\lambda = 0.033$ |
| Parsing Time $T_P$ | Exponential distribution, Mean = 0.13 s |

$$\text{pdf: } f(x) = \lambda e^{-\lambda x}, x \geq 0, \lambda = 7.69$$

*Appendix A.3. VoIP Traffic*

Table A3 presents the most important parameters that characterize the VoIP traffic and that were used in the performed simulations. A simple 2-state voice activity model is considered, and the state of the model coder is updated at the speech encoder frame rate $R = 1/T$, where $T$ is the encoder frame period (typically 20 ms).

**Table A3.** VoIP traffic parameters.

| Parameter | Statistical Characterization |
|---|---|
| Codec | RTP AMR 12.2, Source rate 12.2 kbps |
| Encoder frame length | 20 ms |
| Voice activity factor | 50% |
| SID (Silence Insertion Descriptor) payload | Modelled: 15 bytes (5 bytes + header) SID packet every 160 ms during silence |
| Protocol overhead with compressed header | 10 bits + padding (RTP-pre-header) 4 bytes (RTP/UDP/IP), 2 bytes (RLC/security), 16 bits (CRC) |
| Total voice payload on the air interface | 40 bytes (AMR 12.2) |

*Appendix A.4. Video Streaming Traffic*

The frames of a video stream arrive at regular time interval $T$ correlated with the number of frames per second. Each frame is composed of a fixed number of slices, each slice transmitted as a single packet. The sizes of the slices are modeled by a Truncated Pareto distribution. The video encoder introduces delays between the packets of a frame. These delays are also modeled by a truncated Pareto distribution. The following distributions assume a source video rate of 64 kbps. The video streaming traffic parameters are given in Table A4.

**Table A4.** Video streaming traffic parameters.

| Parameter | Statistical Characterization |
|---|---|
| Inter-arrival time between the beginning of each frame | Deterministic 100 ms (based on 10 frames per second) |
| Number of packets (slices) in a frame | Deterministic, 8 packets per frame |
| Packet (slice) size | Truncated Pareto distribution Mean = $m$ = 20 bytes, Maximum = 250 bytes (before truncation) |
| | pdf: $f(x) = \dfrac{\alpha \cdot k^{\alpha}}{x^{\alpha+1}}, k \leq x < m$, $f(x) = \left(\dfrac{k}{m}\right)^{\alpha}, x = m$, $\alpha = 1.2, k = 10$ bytes |
| Inter-arrival time between packets (slices) in a frame | Truncated Pareto distribution Mean = $m$ = 6 ms, Maximum = 12.5 ms (before truncation) |
| | pdf: $f(x) = \dfrac{\alpha \cdot k^{\alpha}}{x^{\alpha+1}}, k \leq x < m$, $f(x) = \left(\dfrac{k}{m}\right)^{\alpha}, x = m$, $\alpha = 1.2, k = 2.5$ ms |

*Appendix A.5. Interactive Real-Time Services: Online Gaming*

It is supposed that the starting time of an online gaming session is uniformly distributed within [0, 40 ms]. This is necessary for the simulation of the random timing relationship between packet arrival and uplink frame boundary. A maximum delay of 160 ms is allowed for all uplink packets, and a packet is dropped if this maximum delay is not fulfilled. The delay of a dropped packet is counted as 180 ms.

If the average packet delay is greater than 60 ms the gaming session is considered to be in outage. The online gaming traffic parameters are given in Table A5.

**Table A5.** Online gaming traffic parameters.

| Parameter | Statistical Characterization |
|---|---|
| Initial packet arrival | Uniform Distribution<br>pdf: $f(x) = \dfrac{1}{b-a}, a \le x \le b$  $a = 0, b = 40$ ms |
| Packet arrival | uplink: deterministic, 40 ms<br>downlink: Largest Extreme Value distribution<br>pdf: $f(x) = \dfrac{1}{b} e^{-\frac{x-a}{b}} e^{-e^{-\frac{x-a}{b}}}$, $a = 55$ ms, $b = 6$ ms |
| Packet size | Largest Extreme Value distribution<br>pdf: $f(x) = \dfrac{1}{b} e^{-\frac{x-a}{b}} e^{-e^{-\frac{x-a}{b}}}$<br>uplink: $a = 45$ bytes, $b = 5.7$ bytes<br>downlink: $a = 120$ bytes, $b = 36$ bytes |
| UDP header | Deterministic (2 bytes). |

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
