# Peer review of "Game Theory-Based Load-Balancing Algorithms for Small Cells Wireless Backhaul Connections†"

_applsci, doi:10.3390/app13031485_

Round 1

Reviewer 1 Report

General comments: formatting and language

Please note the below comments should be carefully addressed through out the entire paper. The below are just examples:

1.       It is better to use initial small letters for the acronyms, for example: Fuzzy Neural Network (FNN)à fuzzy neural network (FNN).

2.       In general, try not to use Capitalization unless there is a need (e.g. well-known names like Nash)

3.       Double check this point: Define the acronym the first time they appear, then use the acronym of the full definition).

4.       The paper needs careful proof reading, for examples,

5.       Line 13 AP should be APs

6.       -in the abstract, line 23: use should be uses

7.       31, 32 AP should be Aps

8.       223 while fulfilling-à, while fulfilling

9.       224 which returnsà , which returns

Line  -- the expectance—the expectation

Paper formatting should be improved. All equations have very small subscripts and superscripts that make them unreadable, especially if you print them on a paper. Please improve  the equations clarity.  E.g., j in Eq 3 and 4!!!

Nash equilibrium in 393 is in italic, on other locations it is not italic, Please be consistent.

487 algorithm à algorithm,

489: and execute-à and executing

552--- threshold-à threshold

558 flow fi the tunnel à flow fi of the tunnel

Technical comments

1.       Line 249: the main analysis is since you have a Markov model that is valid only if you have a Poison distribution for the data flow. Can you provide a reference for this claim to be added to the lines 249 where you have mentioned this assumption.

2.       Eq 9, what is meant by …., you should be an upper limit for n. The same for m in 547

3.       313 what is condition 4?

4.       Line 350: “The users’ clustering algorithm could be a decentralized one, when no central control unit is employed, or could be a centralized one when an AP Controller module is  used”.

5.       Can you please explain how?

6.       Line 356:  each AP selecting a number of users that can be 356 served with the resources available on the allocated virtual tunnels.  

7.       Explain how?

8.       385: I do not think this is a definition, it should be a theorem.

9.       420: explain how the equation was a result of the method of moments?

10.   423: explain why NE is worse than the optimal?

11.   426: In the following, I was expected to see an equation, where it has a left and right hand variables, can you check Eq 20 .

12.   431: if the packet change the channel, then you may have Jitter, so how you will deal with Jitter? Can you please mention this point and mention the remedy for it.

Author Response

Response to Reviewer 1 Comments

The authors thank the reviewer for the effort and time invested in carrying out the review and for the observations and suggestions made.

Pont 1

- It is better to use initial small letters for the acronyms, for example: Fuzzy Neural Network (FNN)à fuzzy neural network (FNN).

- In general, try not to use Capitalization unless there is a need (e.g. well-known names like Nash)

- Double check this point: Define the acronym the first time they appear, then use the acronym of the full definition).

- The paper needs careful proof reading, for examples,

- Line 13 AP should be APs

- in the abstract, line 23: use should be uses

- 31, 32 AP should be Aps

- 223 while fulfilling-à, while fulfilling

- 224 which returnsà , which returns

- Line-- the expectance—the expectation

Response: Corrections were done

Point 2. Paper formatting should be improved. All equations have very small subscripts and superscripts that make them unreadable, especially if you print them on a paper. Please improve  the equations clarity.  E.g., j in Eq 3 and 4!!!

Response: Unfortunately, the paper template doesn’t specify the exact formatting of the equations. For more complex equations the MathType tool was used with characters formatted to 10pt Palatino Linotype while for the subscript/superscript sub-subscript/superscript the default MathType formatting was used, i.e., 58% subscript/superscript and 42% sub-subscript/superscript. As far as we understand if accepted the paper will undergo a professional editing according to the exact template of the journal. We can change the formatting of the equations, but probably they will be changed again. Some corrections were performed, and the size of the equations was increased.

Point 3.

- Nash equilibrium in 393 is in italic, on other locations it is not italic, Please be consistent.

- 487 algorithm à algorithm,

- 489: and execute-à and executing

- 552--- threshold-à threshold

- 558 flow fi the tunnel à flow fi of the tunnel

Response: corrections performed

Technical comments

Point 4. Line 249: the main analysis is since you have a Markov model that is valid only if you have a Poison distribution for the data flow. Can you provide a reference for this claim to be added to the lines 249 where you have mentioned this assumption.

Response: A new reference, [33], was included in the reference list and mentioned at lines 263-269. A short discussion related to this issue is also included at the mentioned lines.

Point 5. Eq 9, what is meant by …., you should be an upper limit for n. The same for m in 547

Response: if the queue is in equilibrium nj will be limited to a maximum value, in the sense that the fraction of packets leaving behind nj packets in the queue is decreasing with the value of nj, and will approach zero for nj large. In the case of Algorithm A3 it is about the time moments when the algorithm is executed, when a new packet arrives, or when a timer expires.

Point 6. 313 what is condition 4?

Response: occupation rate of the tunnel to be smaller than 1; eq. 4 was changed

Point 7. Line 350: “The users’ clustering algorithm could be a decentralized one, when no central control unit is employed, or could be a centralized one when an AP Controller module is  used”. Can you please explain how?

Response: the explanation is in the text – lines 368 - 373. The APs in discussion, for example the APs deployed in a train or in a building are part of the network of the train (possibly a wired network) or of the building and can communicate and exchange information between them or with some other network elements.

Point 8. Line 356:  each AP selecting a number of users that can be 356 served with the resources available on the allocated virtual tunnels. Explain how?

Response: Let suppose that a single AP is deployed, and 4 virtual tunnels are available for backhaul connection, for example established in 4 different macro cell networks, controlled by different operators. The single AP is performing LB on the 4 tunnels. If 2 APs are deployed each of them will manage, for example, 2 out of the 4 tunnels. The number of tunnels is fixed (the tunnels are leased from the macro cell operators by the train or building operator), the users being also slit between the 2 APs. The use of more APs has the goal of controlling the user-AP links and provisioning of QoS on these links, but the total output capacity remains the same.

Point 9. 385: I do not think this is a definition, it should be a theorem.

Response: changed

Point 10. 420: explain how the equation was a result of the method of moments?

Response: it is supposed that the mixture of traffic is Poisson distributed, the mean value of the distribution is measured experimentally and is equated with the theoretical mean of the Poisson distribution. In this simple case we have only one parameter of the distribution.

Point 11. 423: explain why NE is worse than the optimal?

Response: due to the lack of coordination among the players, when a selfish game is used, the solution obtained at Nash Equilibrium is not the optimal one

Point 12. 426: In the following, I was expected to see an equation, where it has a left and right hand variables, can you check Eq 20 .

Response: solved

Point 13. 431: if the packet change the channel, then you may have Jitter, so how you will deal with Jitter? Can you please mention this point and mention the remedy for it.

Response: a comment was inserted in the text regarding the large jitter which can be generated by the selfish algorithm. This is another reason why not using it. If a timer is used and the LB process is performed only when this timer expires the amount of jitter generated will decrease and can be compensated by the buffering used at the receiver – see lines 455-458

Reviewer 2 Report

The paper considers the situations when small cells are deployed temporarily or are deployed in a vehicle transporting a large number of passengers, situations when the traditional wired or wireless backhaul solutions could be too costly to be used. The paper proposes, as an alternative solution, the use as backhaul connection the aggregation of the wireless links set up in the macro cells which cover the location of the small cell. But the results need more experiments.

Author Response

Response to Reviewer 2 Comments

The authors thank the reviewer for the effort and time invested in carrying out the review and for the observations and suggestions made.

Pont 1. The paper considers the situations when small cells are deployed temporarily or are deployed in a vehicle transporting a large number of passengers, situations when the traditional wired or wireless backhaul solutions could be too costly to be used. The paper proposes, as an alternative solution, the use as backhaul connection the aggregation of the wireless links set up in the macro cells which cover the location of the small cell. But the results need more experiments.

Response: The observation is perfectly valid, more experiments substantiating better the claimed results. On the other hand, the performance of any load balancing algorithm depends strongly on the type of the user flows, the distribution of these flows and the relation between the requested transfer rate and offered capacity. If the requested transfer rate is significantly smaller than the offered capacity it doesn’t matter what load balancing algorithm is used, the tunnels being capable to accommodate easily the users’ flows. If the requested transfer rate is larger than the offered capacity again it doesn’t matter what algorithm is used. The difference between the various LB algorithms can be noticed only when the requested total rate is close to the offered total capacity and an optimal distribution of the flows are required to fill the tunnels. The selected traffic mixture represents such a situation. These explanations were included in the text in the Simulation results and Discussions section - lines 1005-1010. A short discussion about the selection of the traffic mix used in simulations already was included in the paper: lines 913-921. If the simulations are performed with other traffic mixtures and/or tunnel parameters in the above-mentioned conditions, i.e., similar level of congestion, the results will be similar, because the level of congestion is the main parameter which requires more effective solutions – in our case the Combinatorial auction algorithm, followed by the “simple” Auction based algorithm.

In what concerns the clustering algorithms, their performance is strongly related to the distribution of the users and the type of traffic generated by the users. The results included in the paper are among the most relevant ones obtained.

Reviewer 3 Report

The paper presents several algorithms for balancing the load of multiple flows across a few independent channels (tunnels). The algorithms span multiple approaches (greedy, game theory, auction, clustering, round robin, etc.) and variants (cooperative, adversarial). After introducing the different algorithms, it analyzes the distribution of packet latency in multiple scenarios using sythetic traffic and network models.

The paper has some English mistakes, but it is reasonably well written for the most part. The introduction includes a large number of references to related work, the algorithms are clearly presented, and their performance is analyzed through extensive simulations.

My main concern with the paper is that, although it seems correct and reasonably well written, I do not see much in terms of new ideas or results that could be of interest to the research or industry community. The proposed algorithms are just variants of existing ideas and the performance of the algorithms is as expected (clustering users to balance traffic reduces delay, cooperation also helps, centralized algorithms outperform distributed greedy ones, etc.) The absolute numbers in the results are not useful either, since they just apply to the specific scenario being simulated. The algorithms do not seem particularly insightful...

If the paper were to be accepted, here are a few major comments that I think should be addressed before publication:

1.- The introduction mentions several existing works. The paper should mention whether they could be applied to solve the problem under consideration and, if so, why their proposed algorithms are better.

2.- The first 3 paragraphs in section 2.1 basically repeat the same as in the introduction. I propose merging them.

3.- I find that there are too many algorithms being proposed/described (There are 7 "algorithm" blocks in the paper, and some of them have multiple variants). Furthermore, the choices performed in the algorithms are not justified. They are not  The paper should make it very clear which one is the one that it is proposing should be used. If different algorithms are proposed for different scenarios, that should also be clarified.

4.- The selfish routing and the auction algorithm seem to be very similar: whenever a packet arrives, they evaluate the latency on each tunnel and pick the lowest, right? The main difference resides in the penalty to change tunnels that the latter introduces.

5.- The parameter and function choices in the proposed algorithms seem rather arbitrary, not optimized. For example, it seems like the choice of the amount in the bids (Eq 22) and the potential functions (Eq 26) were chosen using heuristic logic, not mathematical optimization. I suggest that the authors present a mathematical argument or at least some sort of justification for their choices. 

6.- Would it be possible to find a globally optimal solution, maybe using a branch and bound or exhaustive search method? I understand that it would be too computationally expensive for a practical scenario, but it might be useful to include it in the results so as to see the gap to optimality for each of the algorithms.

Minor comments:

2.1.1 Is the function Tj(nj, Cj(t)) convex or linear?

Eqs should be properly punctuated and one should not indent the text after an equation unless a new paragraph is really starting (see eg. after Eq 18) 

There are occasional unjustified changes in the font, eg. after Eq 13 and first sentence in 2.2

First sentence in 2.1.2 should be "In some circumstances there could be several small cells available to which..."

In 2.2, is the bold K in line 362 the same as the non-bold K in 359? Be consistent

The point of reference to [37] in 2.2.1 is unclear

Define "social cost"

Several pages are devoted to describing the statistical characteristics of different traffic patterns (3.1.1). I suggest moving those to an appendix section.

Many result figures include both the pdf and the cdf of the data. Is there a point to include both?

Instead of writing "In Fig. X it is presented" write "Fig X presents"

Author Response

Response to Reviewer 3 Comments

The authors thank the reviewer for the effort and time invested in carrying out the review and for the observations and suggestions made.

Point 1. The introduction mentions several existing works. The paper should mention whether they could be applied to solve the problem under consideration and, if so, why their proposed algorithms are better.

Response: the introduction shows that GT based algorithm are good for solving various load/traffic/resource optimization problems. The load balancing problem considered is in fact a resource optimization problem which can be solved in many ways, off course. The paper proposes such game theory based optimization algorithms which in fact are variations of basic game types, like selfish games or auction based games. What is different in our case is the way the game is defined in the specific cases considered, the parameters involved, and the quality indicators targeted. So, the response to the question if other GT based algorithms proposed in the papers from the reference list can be used to solve our problem is YES, BUT not without adapting them to the specific problem in question with specific objectives and constraints, and this is practically the contribution of the paper. Some comments were introduced in the Introduction section – see lines 168-172, 183-190

Point 2. The first 3 paragraphs in section 2.1 basically repeat the same as in the introduction. I propose merging them.

Response: if necessary, we can do this, but a separate section for problem formulation and modeling is often used and such section can present the goals of the paper more clearly as well as the basic modeling issues. In order to avoid repetitions the last part of the introduction section was modified – lines 176-181 deleted. Lines 204-211 were deleted also from the Problem Formulation section

Point 3. I find that there are too many algorithms being proposed/described (There are 7 "algorithm" blocks in the paper, and some of them have multiple variants). Furthermore, the choices performed in the algorithms are not justified. They are not  The paper should make it very clear which one is the one that it is proposing should be used. If different algorithms are proposed for different scenarios, that should also be clarified.

Response: Yes, there are in total 7 algorithms, but there are proposed 3 algorithms for load balancing in increasing order of complexity, practically a selfish game and two versions of the auction based games. The 2 reference algorithms are just used to assess the performance improvements of the game theory based algorithms – they are not part of the proposed algorithms. The Round Robin algorithm is practically the situation with a simple multipath routing, while the multiple knapsack was considered because is a basic solution which tries to fill the tunnels (knapsacks) with flows (items). Of course, many other solutions could be considered as reference – the names of the sections describing these algorithms were changed to express clearly that they are just for reference.

The game theory based clustering algorithms are variants of the LB algorithms but their goals are totally different and should be considered separately. They act on the user – AP link in order to provide QoS on this link (necessary for providing end to end QoS) and distribute the users between several installed APs – for example in a long train (like a TGV train with 20 carriages) we cannot expect that a single AP can cover the entire train and the users should be attached to one AP in such a way to provide end to end QoS. In this case we have two variants of the algorithm, with and without cooperation/ coordination, the algorithm without coordination being a selfish type algorithm. So, in total we have 4-5 algorithms (depending how we are considering the auction based algorithms) which doesn’t seems too much.

In the results section the performances of the 2 groups of algorithms (acting on the tunnel respectively on the access link) were analyzed separately and advantages / disadvantages of each algorithm is discussed. The results discussion and analysis section, Section 3.3.1, was extended with some supplementary comments in order to express more clearly what algorithm should be used and when – see lines 1011-1018.

Point 4. The selfish routing and the auction algorithm seem to be very similar: whenever a packet arrives, they evaluate the latency on each tunnel and pick the lowest, right? The main difference resides in the penalty to change tunnels that the latter introduces.

Response: not quite true. In the selfish game each player acts selfishly without considering the consequences of its actions on other players, but the algorithm is simple, and no central coordinator is required. In the auction based algorithm each player submits a bid to a contractor, the LB decision module, (a central module) in our case, and the lowest bid is selected. Of course, there are similarities, but the results are different even in the no penalty case – the auction based algorithm is better. See also lines 513-521 for differences.

Point 5.- The parameter and function choices in the proposed algorithms seem rather arbitrary, not optimized. For example, it seems like the choice of the amount in the bids (Eq 22) and the potential functions (Eq 26) were chosen using heuristic logic, not mathematical optimization. I suggest that the authors present a mathematical argument or at least some sort of justification for their choices. 

Response: As it is stated in the paper “Finding the optimal value of the bid is a complex problem that depends on the knowledge of the other bidders and their way of acting.” lines 479-481, the optimal way of generating the bids is a complex problem, beyond what is proposed in this paper. The bid generated according to eq. 21-22 represents the expected delay reported to the average interpacket arrival time (which is 1/ packet rate), which is an estimation of the transmission resources necessary to provide the imposed delay, in other words the bandwidth used, and this is practically the cost necessary to transport the data packets. Some comments were included in the text – lines 485 – 488. The tunnel using the less bandwidth wins the contract. Another component of the price is the number of packets in the queue, meaning the amount of memory necessary. The number of packets in the queue will have effect on the evolution of the delay, and to keep the imposed delay it is possible to be needed more bandwidth. Some comments were included in the text – lines 496 – 499. All the parameters used to generate the bid are easily accessible if the CSI and QSI is available.

The potential functions are a measure of the users’ satisfaction for the two main types of services, real-time and non-real time. In the case of real-time services, a power function having as negative exponent the delay, eq. 24, can express very well the user satisfaction which is decreasing fast if the delay is increasing. Another value of the base can be used as well. Some comments were added to the exiting explanations – lines 544-545. The used power function allows also a good separation between different types of real time flows, see lines 550-552. In the case of best effort traffic, the used potential function, eq. 26, also expresses the user satisfaction with the increasing values of the throughput reported to the expected average rate value. The hyperbolic tangent function it is a good choice for representing a phenomenon with progressive transition between two states: not satisfied at all and totally satisfied (in our case) – comment included in the text, lines 573-576. The hyperbolic tangent function is also very similar to the sigmoid function used with neural networks for its differentiability characteristics.

Point 6. Would it be possible to find a globally optimal solution, maybe using a branch and bound or exhaustive search method? I understand that it would be too computationally expensive for a practical scenario, but it might be useful to include it in the results so as to see the gap to optimality for each of the algorithms.

Response: the optimization problems presented in the manuscript are NP problems and for this finding the globally optimal solution with a deterministic machine is impossible. The optimality gap has sense only for the Combinatorial auction based algorithm which is clearly the best among the considered algorithms. The closest to the optimal solution would be provided by a linear program that solves the flow allocation on tunnels while tunnel occupation rate constraints should be fulfilled, and some objective functions should be maximized – see the system response function defined at lines 378-385. But the flow rates are not constant, at least not for all flows, and the tunnel capacities are not constant, and the optimal allocation should be recomputed according to the burstiness of the flows and variation of tunnels’ capacities. Considering only the average flow rate is not enough – see the performance loss of the Multiple Knapsack algorithm. Practically the Combinatorial Auction algorithm it is doing this (it is an alternative to linear programing but could be a form of linear programing – relation between linear programing and game theory is quite complex), the QoS parameters of the flows are checked by computing the potential functions and when significant changes of the QoS is detected the flows are reallocated. We can expect that the Combinatorial Auction algorithm to be close to an optimal solution. Explanations concerning the differences between the performance of the considered algorithms are already in the text - see lines 933-976 and 977-1004.

Just to show that finding the optimal solution is NP hard, in the case of the clustering algorithms, we tried to solve the problem stated by eq. (28) using linear programing and the Matlab solver. The simulation of a 10 second time period took an entire night on a good computer.

Also, the gap between the non-cooperative and cooperative GT based clustering algorithms is relatively small and we estimate that an optimal solution will be close to the performance of the cooperative clustering algorithm.

Point 7. Minor comments:

2.1.1 Is the function Tj(nj, Cj(t)) convex or linear?

Response: Convex – see line 258

Eqs should be properly punctuated and one should not indent the text after an equation unless a new paragraph is really starting (see eg. after Eq 18) 

Response: corrected

There are occasional unjustified changes in the font, eg. after Eq 13 and first sentence in 2.2

Response: corrected

First sentence in 2.1.2 should be "In some circumstances there could be several small cells available to which..."

Response: corrected

In 2.2, is the bold K in line 362 the same as the non-bold K in 359? Be consistent

Response: non bold K is a number, bold K is a set

The point of reference to [37] in 2.2.1 is unclear

Response: reference removed

Define "social cost"

Response: The penalties suffered by the players. The best configuration would be the social optima, each player gains as much as possible

Several pages are devoted to describing the statistical characteristics of different traffic patterns (3.1.1). I suggest moving those to an appendix section.

Response: the traffic modeling was moved in Appendix A

Many result figures include both the pdf and the cdf of the data. Is there a point to include both?

Response: If necessary, the pdf figures can be removed, but they give more insights on the probabilistic process. Not being any page limitations, we consider that can be kept both types of figures.

Instead of writing "In Fig. X it is presented" write "Fig X presents"

Response: corrected

Round 2

Reviewer 1 Report

The authors have addressed most of my comments. However, I recommend that they reflect their explanation to me to the paper (they did it for some of my comments). As the reader may have the same doubts I had when I read the paper.

Author Response

Pont 1. The authors have addressed most of my comments. However, I recommend that they reflect their explanation to me to the paper (they did it for some of my comments). As the reader may have the same doubts I had when I read the paper.

Response

Some of the answers/explanations to the reviewer’s comments were already included in the text, and other answers/explanations were included in the new version – see the details below

Technical comments at round 1

Point 4. Line 249: the main analysis is since you have a Markov model that is valid only if you have a Poison distribution for the data flow. Can you provide a reference for this claim to be added to the lines 249 where you have mentioned this assumption.

Response: explanations already included in the text, see lines 264-270

Point 5. Eq 9, what is meant by …., you should be an upper limit for n. The same for m in 547

Response:– new explanations were inserted at lines 304-306 and some changes were performed at line 594

Point 6. 313 what is condition 4?

Response: eq. 4 was changed in the previous version of the manuscript

Point 7. Line 350: “The users’ clustering algorithm could be a decentralized one, when no central control unit is employed, or could be a centralized one when an AP Controller module is   used”. Can you please explain how?

Response: the explanations are already in the text – lines 372 - 377.

Point 8. Line 356:  each AP selecting a number of users that can be 356 served with the resources available on the allocated virtual tunnels. Explain how?

Response: new explanations were included in lines 380-383

Point 9. 385: I do not think this is a definition, it should be a theorem.

Response: changed in the previous version of the manuscript

Point 10. 420: explain how the equation was a result of the method of moments?

Response: explanations are already in the text – lines 444-449; one line was added

Point 11. 423: explain why NE is worse than the optimal?

Response: some new explanations and rephrasing are done at lines 450-454

Point 12. 426: In the following, I was expected to see an equation, where it has a left and right hand variables, can you check Eq 20 .

Response: solved in the previous version

Point 13. 431: if the packet change the channel, then you may have Jitter, so how you will deal with Jitter? Can you please mention this point and mention the remedy for it.

Response: explanations are already in the text, see lines 466-468

Reviewer 3 Report

The authors have answered most of my questions satisfactorily, incorporated many of my suggestions, and improved the writting significantly. However, many equations are still missing punctuation (e.g. 10 and 13 should have a period at the end) and many English mistakes still remain (e.g. line 415 should be "according to Little's law", line 482 "It should be noted" and line 497 "is equivalent to minimization")

Also, I still feel that the paper does not contribute much in terms of new ideas or results that could be of interest to the research or industry community. It seems to be adapting some existing ideas to incorporate some additional constraints on the problem and then analyzing the results in a few particular instances.

Author Response

Pont 1. The authors have answered most of my questions satisfactorily, incorporated many of my suggestions, and improved the writting significantly. However, many equations are still missing punctuation (e.g. 10 and 13 should have a period at the end) and many English mistakes still remain (e.g. line 415 should be "according to Little's law", line 482 "It should be noted" and line 497 "is equivalent to minimization")

Response: the paper was checked and re-checked again and corrections were done. Unfortunately, the paper is relatively long and it is possible that some editing and English grammar issues still remain.

Point 2. Also, I still feel that the paper does not contribute much in terms of new ideas or results that could be of interest to the research or industry community. It seems to be adapting some existing ideas to incorporate some additional constraints on the problem and then analyzing the results in a few particular instances.

Response: the authors agree that the paper doesn’t bring a significant new theoretical contribution, but they believe that the paper does bring contributions in what concerns the practical part (interest to the industry). Some years ago the authors were involved in a FP7-SME project (the UCONNECT project), dedicated to SMEs (small and medium enterprises), which proposed to develop a ubiquitous connectivity system for public transportation, a connectivity system that was using as backhaul connections the links established in several macro cells managed by different operators.  In the project was developed such a connectivity platform (of course a prototype system), which finally worked relatively well. Extensive testing took place in Cluj, Romania, and in Switzerland. The algorithm used for the load balancing operations was a variant of the Multiple Knapsack, and also some MADM algorithms were tested.  

The main issues with the developed Multiple Knapsack algorithm were the measurement of the average flow rate (who is exactly a flow, which is the minimum number of packets involved by a flow) and the measurement of the spare capacity of the macro cell links, which depends strongly on the used RRM algorithms and the policies applied by the network operator. The GT algorithms developed in the paper need only access to the buffers of the IP stack and of the wireless interface, and all the decisions are taken based on delay measurements. It is not really necessary to define a flow and it is not necessary to measure the spare capacity of the links. Overloaded links will have larger delays and more packets in the driver buffer.  We strongly believe that the GT algorithms developed, more exactly the auction based algorithms are very good candidates for implementing a load balancing process over several communication tunnels in real systems, the complexity is limited (or more exactly can be made limited by some changes of the algorithm – for example do not perform the LB operation on every packet) and it is not necessary to use complex CSI acquisition operations. It is only necessary to access the above-mentioned buffers and changing the wireless interfaces will require no changes in the LB decision algorithms. Based on the authors’ knowledge many companies work on similar solutions for various transportation market segments like maritime and aviation using tunnels over satellite networks, and high-speed trains, and public transportation systems using terrestrial networks.

In the case of the clustering algorithms again it is necessary only to provide access to the buffers and only the transmission power is manipulated in order to avoid more complex or non-standard interactions between the APs and the users (which is not possible for COTS terminals), but good results can be obtained even in these circumstances.

Also, should be mentioned that the trends in B5G are to incorporate distributed intelligence in the network, and more and more independent decisions to be taken as close as possible to the edge (native AI). To have devices that are endorsed with independent decision capabilities they have to operate based on knowledge harvested in the local environment. This is what the paper tries to emphasize, what are the possible techniques (even if they are well studied and used in other domains) that allow optimized local decisions without data collected/provided by the standardized network connections (e.g. WiFi network, 3GPP xG tunnels, etc.)

Some comments related to the flexibility of the GT based algorithm and the limited amount of information necessary to perform the LB operation were inserted at lines 1085-1098 and lines 1204-1209.